# Geographic pair matching in large-scale cluster randomized trials

Benjamin F. Arnold [1,2] ✉, Francois Rerolle[1], Christine Tedijanto[1], Sammy M. Njenga[3], Mahbubur Rahman [4], Ayse Ercumen[5], Andrew Mertens [6], Amy J. Pickering [7,8], Audrie Lin [9], Charles D. Arnold[10], Kishor Das [11], Christine P. Stewart [10], Clair Null[12], Stephen P. Luby [13], John M. Colford Jr[6], Alan E. Hubbard [14] & Jade Benjamin-Chung [8,15]

Cluster randomized trials are often used to study large-scale public health interventions. In large trials, even small improvements in statistical efficiency can have profound impacts on the required sample size and cost. Location integrates many socio-demographic and environmental characteristics into a single, readily available feature. Here we show that pair matching by geographic location leads to substantial gains in statistical efficiency for 14 child health outcomes that span growth, development, and infectious disease through a re-analysis of two large-scale trials of nutritional and environmental interventions in Bangladesh and Kenya. Relative efficiencies from pair matching are ≥1.1 for all outcomes and regularly exceed 2.0, meaning an unmatched trial would need to enroll at least twice as many clusters to achieve the same level of precision as the geographically pair matched design. We also show that geographically pair matched designs enable estimation of fine-scale, spatially varying effect heterogeneity under minimal assumptions. Our results demonstrate broad, substantial benefits of geographic pair matching in large-scale, cluster randomized trials.

In cluster randomized trials, investigators randomly allocate groups of individuals to receive an intervention that manipulates the physical or social environment, or cannot be delivered to individuals[1]. Cluster randomized trials are especially common in studies of infectious or contagious outcomes, where estimating the effect of an intervention deployed to groups of individuals is paramount[2]. The number of cluster randomized trials indexed in PubMed has increased almost exponentially from 12 in the year 2000 to 465 in 2022 (Supplementary Fig. 1). Many of the early cluster randomized trials were small, randomizing fewer than 30 units[3], but in recent years many large-scale trials have enrolled hundreds and even thousands of clusters[4–9]. Large-scale cluster-randomized trials have become increasingly common in the Global South to study real-world (pragmatic) interventions with a goal toward improved generalizability in diverse populations[2]. Large trials are often expensive, so methods to improve their efficiency or gain additional insights from them can have an important impact on their

[1]Francis I. Proctor Foundation, University of California, San Francisco, CA, USA. [2]Department of Ophthalmology, University of California, San Francisco, CA, USA. [3]Eastern and Southern Africa Centre of International Parasite Control, Kenya Medical Research Institute, Nairobi, Kenya. [4]Environmental Interventions Unit, Infectious Diseases Division, icddr,b, Dhaka, Bangladesh. [5]Department of Forestry and Environmental Resources, North Carolina State University, Raleigh, NC, USA. [6]Division of Epidemiology, School of Public Health, University of California, Berkeley, CA, USA. [7]Department of Civil and Environmental Engineering, University of California, Berkeley, CA, USA. [8]Chan Zuckerberg Biohub, San Francisco, CA, USA. [9]Department of Biobehavioral Health, Pennsylvania State University, University Park, PA, USA. [10]Department of Nutrition, University of California, Davis, CA, USA. [11]CURAM, SFI Research Centre for Medical Devices, University of Galway, Galway, Ireland. [12]Mathematica, Washington, DC, USA. [13]Infectious diseases and Geographic Medicine, Stanford University, Stanford, CA, USA. [14]Division of Biostatistics, School of Public Health, University of California, Berkeley, CA, USA. [15]Department of Epidemiology and Population Health, Stanford University, CA, USA. ✉e-mail: ben.arnold@ucsf.edu

value. Here, we assess geographic pair matching as a logistically simple design strategy with the potential to increase the value of large cluster-randomized trials through improved efficiency (decreased sample size) and through unique insights from spatially explicit analyses of effect heterogeneity under minimal assumptions.

Random allocation within strata or pair matched clusters can improve statistical efficiency if the variable(s) used to stratify or match are strongly correlated with the outcome and the stratification or matching variables are used in the analysis[3]. Restricted randomization is another strategy sometimes used in trials to ensure balance on baseline covariates, but since the restricted randomization process is generally ignored in the analysis it is unlikely to improve statistical efficiency[3]. Pair matching is an extreme form of stratification whereby investigators first create pairs of clusters that are similar by some measure, such as the baseline outcome, and then randomly allocate one cluster to receive an intervention and the other to serve as a control. The main benefits of pair matching are that it ensures balance on characteristics used to match and that it can increase statistical power if trial outcomes are highly correlated within matched pairs and the analysis accounts for the matched design.

In large, cluster randomized trials, the decision of whether to pair match or stratify the randomization can have profound consequences for the conduct, power, and cost of the trial. Yet, there are mixed recommendations about pair matching in the methods literature. Early methodologic studies generally recommended against pair matching on the basis of analytic drawbacks and loss of statistical power through fewer degrees of freedom in the analysis, which can be an issue in small trials with fewer than 20 clusters[10–13]. Later methodologic contributions drew on empirical examples from economic outcomes in Mexico's Seguro Popular public insurance program, which used a large cluster-randomized, pair matched design (74 matched pairs), and argued strongly for use of baseline characteristics to pair match randomization based on considerations of statistical efficiency and robustness to unexpected events, such as co-interventions or epidemics[14–16].

A major logistical concern for pair matching in large-scale trials is the availability of baseline information used to pair match at the time of randomization[2]. Any potential savings through reduced sample size due to pair matching could be offset by costs imposed by obtaining the information used to match. In large-scale trials, enrollment and baseline measurements could require months or even years to complete, and it is often impractical to delay randomization and treatment until the relevant information is available to match. Even if done, gains in statistical efficiency are not guaranteed from pair matching. For example, members of our team conducted a cluster randomized trial of household water treatment in rural Bolivia in which 22 communities were pair matched by the baseline incidence of the primary outcome, diarrhea[17], only to find afterward that baseline incidence was uncorrelated with incidence during the trial (and thus would not have improved efficiency)[18].

Geographic location is one characteristic, known in advance, that can be used to pair match and randomize clusters in real-time, without the need to census the entire study population before randomization. Location integrates many complex environmental and socio-demographic characteristics that often play a central role in human health and wellbeing[19], so pair matching by location could balance myriad population characteristics, many that are unknown or unmeasurable. As field teams progress through enrollment, geographically proximate clusters can be paired and randomized. If a trial uses rolling enrollment with geographic pair matched randomization, the design also pair matches on calendar time — an important consideration for seasonally varying outcomes or interventions. Geographic pair matching ensures that intervention and control groups are evenly balanced over a study region, which means that any independent programs or policy changes that could influence a portion of the study population defined by geography are evenly balanced between groups — one such co-intervention overlapped one of the trials we study here[20]. Despite the potential value of geographic pair matching for cluster randomized trials, to our knowledge there have been no studies of its potential cost or benefit in the context of large-scale epidemiologic field trials.

Here, we assessed the potential benefit of geographic pair matching in cluster randomized trials, using empirical results from two trials of improved nutrition and WASH (water, sanitation, and hand-washing) interventions in Bangladesh and Kenya: the WASH Benefits study[4,5,21]. The trials were large in scale (>700 clusters each), used geographic pair matching in the design, and measured a broad set of health and development outcomes among children born into study compounds. Our main objective was to assess the effects of geographic pair matching on statistical efficiency across 14 outcomes spanning child growth, infectious disease, and cognitive development. We compared efficiency gains from geographic pair matching with other randomization strategies and studied key drivers of efficiency gains with respect to outcome prevalence, outcome clustering, and the trials' spatial scale to provide insight into the design of future large-scale trials. Additionally, we illustrate how geographic pair matching enables estimation of effect heterogeneity by geographically continuous modifiers under minimal assumptions, which can be used to extrapolate trial effects to adjacent populations or help inform targeting of future health interventions.

## Results
### WASH Benefits study design, study population, and setting
The WASH Benefits study included two cluster-randomized trials conducted between 2012-2016 in Bangladesh and Kenya that enrolled pregnant mothers and their newborn children[4,5,21]. Study populations were rural and selected to be representative of areas of each country with high burdens of child growth faltering and diarrhea (Fig. 1a). Between 8-12 geographically proximate household compounds with pregnant mothers were grouped into clusters, and 8 (Bangladesh) or 9 (Kenya) contiguous clusters were grouped into blocks for randomization. In each matched block, two clusters were randomized to control, and six clusters were randomized to different interventions (Fig. 1b). The Kenya trial included one additional cluster in each block for a passive control that included no visits beyond measurement (Methods). Interventions included household improvements for chlorine-based water treatment (W), compound-level sanitation (S), household handwashing with soap (H) or their combination (WSH). Additional groups received a nutritional intervention (N) or a combined package of all interventions (WSH + N, details in Methods). Newborn children were followed for two years, with outcomes measured at approximate ages 12 and 24 months.

The trials technically matched 8-tuples but for clarity of exposition in the present analysis, we limited the study population to control and nutritional intervention clusters (N, WSH + N), and pooled outcomes within each group to create a simplified, balanced design with which to focus on the methodologic aspects of geographic pair matching (Fig. 1c). The nutritional intervention promoted exclusive breastfeeding through age six months and continued breastfeeding through age 24 months plus complementary feeding practices and, from ages 6-24 months, a daily small-quantity lipid based nutrient supplement (SQ-LNS) that included micro- and macronutrients. In Bangladesh, this included 90 matched pairs (360 clusters) and in Kenya this included 72 matched pairs (288 clusters). We included in the analysis 14 previously published endpoints from the trials that spanned child growth[4,5], child development[22,23] and infectious disease[4,5,24–26] and captured a broad variety of outcome types that might be measured in large-scale field trials (Supplementary Tables 1, 2). Sample sizes were similar within pairs in Bangladesh (Supplementary Fig. 2) and Kenya (Supplementary Fig. 3), with slightly more variability in Kenya.

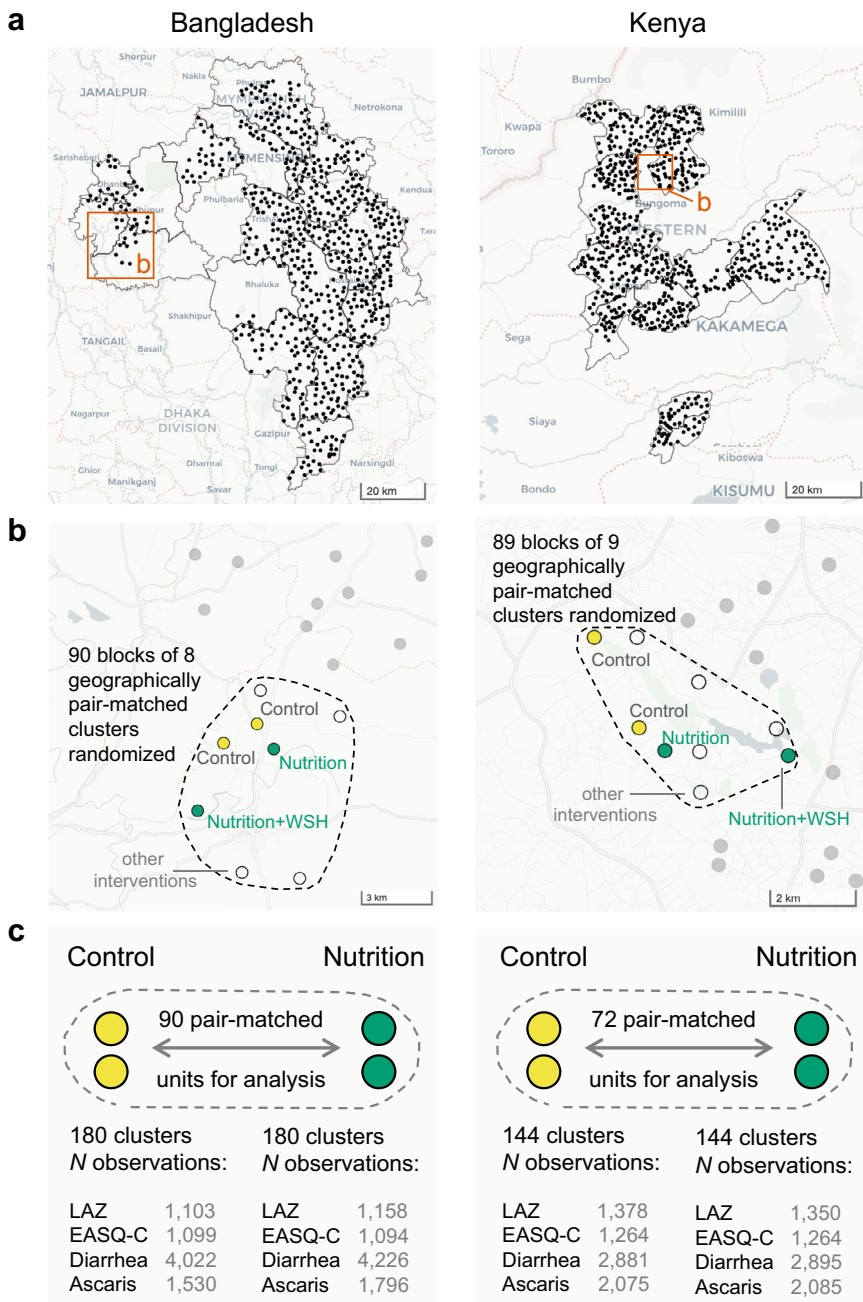

**Fig. 1 | Overview of geographically pair matched designs in the WASH Benefits Bangladesh and Kenya trials. a** Large-scale cluster randomized trials in Bangladesh (n = 720 clusters) and Kenya (n = 702 clusters). Points indicate study clusters, with subdistricts and panel b insets outlined. **b** Clusters were geographically pair matched in blocks of 8 (Bangladesh) or 9 (Kenya) and then randomized. Children in the control and nutritional intervention clusters were included in the present analyses. **c** Sample sizes included in analyses for four representative outcomes, length-for-age z (LAZ), verbal communication scores (EASQ-C), diarrhea and *Ascaris* sp. infection. The Kenya trial was restricted from 89 to 72 blocks with a balanced set of control (2) and nutrition (2) clusters. Supplementary Information Tables 1 and 2 include sample sizes for all outcomes. Open source data from OpenStreetMap with rendering from CARTO using R's leaflet package. Created with notebook https://osf.io/bzrpk.

## Efficiency gains resulting from geographic pair matching

A common parameter of interest in a trial is the mean difference between the intervention and control groups, which is often called the Average Treatment Effect (ATE). The relative efficiency of a pair matched design compared to an unmatched design can be defined as the ratio of the variances of the ATE of the unmatched to pair matched design. Ignoring pairs in the analysis of a pair matched design is an unbiased estimate of the variance under an unmatched design[15]. We could therefore estimate the observed relative efficiency by comparing the variance of the unmatched versus pair matched analyses

(Methods). We additionally approximated relative efficiency gains from geographic pair matching using outcome correlation between matched pairs. In its simplest form, relative efficiency reduces to a function of the outcome correlation between matched pairs (details in Methods)[15,27]: $Reff = (1 - r)^{-1}$, where $r$ is the correlation between matched pairs. For $r = 0.5$, the predicted relative efficiency is 2, suggesting that an unmatched trial would need to enroll twice as many clusters as a pair matched trial to achieve the same level of precision. We estimated mean outcomes at the cluster level, and then estimated the correlation between cluster-level outcomes within matched pairs,

noting that observed outcomes reflect correlation (if any) induced by matching as well as the treatment effect, which may be heterogeneous across pairs.

Geographic pair matching led to pair-wise outcome correlations that ranged from 0.15 to 0.65 in Bangladesh and 0.05 to 0.70 in Kenya, with corresponding relative efficiency gains that were substantial, ranging from 1.1 to 3.3 (Fig. 2a). Outcome rank and magnitude of pairwise correlation differed between countries. In Bangladesh, child growth outcomes had high pair-wise correlation and child development outcomes much lower correlations, while the reverse was generally observed in Kenya. In both countries, particularly large efficiency gains were observed for *Ascaris* sp. and *Giardia* sp. infections and length-for-age z scores.

Observed efficiency gains that compared the variance of the pair matched estimator with a difference in means estimator that ignored pair matching were very close, albeit slightly larger, than gains predicted based on weighted correlation of pair-wise outcomes (Fig. 2b), suggesting that the approximate relationship between correlation and relative efficiency worked well in these trials.

Correlations that weighted pairs by their sample size were as high or higher than unweighted estimates, sometimes substantially higher in the Kenya trial (Supplementary Fig. 4a), consistent with a previous analysis of economic outcomes[15]. Unlike the close relationship between observed and predicted relative efficiency using weighted correlation (Fig. 2b), observed relative efficiency gains were generally larger than predicted based on unweighted correlation (Supplementary Fig. 4b). This result reinforces the increased efficiency of analyses that weight by cluster size even when cluster sizes are relatively similar and independent of the outcome.

## Outcome characteristics and efficiency gains from geographic pair matching

Given the observed differences in efficiency gains between outcomes and countries, we sought to identify underlying features of the trials or outcomes that could help explain the observed differences. We focused on three features related to the spatial aggregation and variation in outcomes: outcome intra-cluster correlation (ICC), outcome spatial autocorrelation measured by Moran's I, and outcome prevalence for binary, infectious disease outcomes. We estimated the ICC, Moran's I, and prevalence using only control group clusters to avoid the potential influence of intervention.

There was a positive relationship between efficiency gains and outcome ICC, spatial autocorrelation, and outcome prevalence (Supplementary Fig. 5). The results help illustrate that spatial variation in the outcome underlies efficiency gains achieved through geographic pair matching, but substantial variation in the relationships suggest that there are unique features of geography that contribute to efficiency gains that these summary statistics fail to capture.

## Trial size and efficiency gains from geographic pair matching

Trials that enroll clusters over larger geographies might include more variation in outcomes through inclusion of more diverse populations or environmental characteristics. If true, we reasoned that pair-wise outcome correlation, and thus relative efficiency gains, would be higher for trials with larger geographic footprints. To study the effect of trial size, we conducted a resampling study that sampled with replacement geographically proximate matched pairs for trial sizes ranging from 10 to the maximum number of pairs in each country, 90 in Bangladesh, 72 in Kenya. The resampling approach held cluster sizes fixed at their actual size, so resampled trials varied in size according to the number of matched pairs. For each trial size, we estimated the mean pair-wise outcome correlation and corresponding relative efficiency across replicates. The specific relationships estimated through the approach are empirical and particular to each outcome and trial.

Relative efficiencies > 1 were observed for nearly every outcome at any trial size, even as small as 10 geographically contiguous matched pairs (Fig. 3, with underlying correlation and uncertainty in Supplementary Figs. 6, 7). Study clusters were separated by ≥1 km to prevent spillover effects, so the number of proximate matched pairs roughly corresponds with the maximum distance between pairs in km in both trials, as a measure of the simulated trials' geographic footprint (Supplementary Fig. 8). Outcomes with greatest gains in relative efficiency at larger trial sizes were those with greater between cluster variance and spatial autocorrelation. For example, in Bangladesh outcomes with highest gains in relative efficiency had high ICCs: *Ascaris* sp. (ICC = 0.05), *Giardia* sp. (0.07), length-for-age z (0.06), weight-for-age z (0.07) and *Trichuris* sp. (0.05) and similarly high values of Moran's I (Supplementary Table 1). Patterns were similar for Kenya (Supplementary Table 2) and reinforce the positive relationship between efficiency gains and between-cluster variability and spatial autocorrelation (Supplementary Fig. 5). Given total variance is fixed, the results suggest more effective geographic matches capture a greater degree of outcome variation between clusters when there is greater between-cluster variance and spatial autocorrelation in the outcome.

## Geographic pair matching versus stratification or pair matching on primary outcomes

Geographic pair matched randomization results in a very fine stratification by geography. An intermediate design strategy, often favored in guidance on the design of cluster randomized trials[3,11,12], is to stratify randomization by geographic areas such as district or subdistrict, with the rationale that the stratification would still balance groups by geographic characteristics but would use fewer degrees of freedom and allow for greater flexibility in the analysis. We re-analyzed the trials, stratifying the analysis by subdistrict-level administrative units in each country − 19 *zillas* in Bangladesh, and 10 sub-counties in Kenya (shown in Fig. 1a). This alternate analysis mimics a design with stratified randomization, albeit optimistically since pair matching ensures a more even spatial distribution of control and intervention clusters than would be expected by chance under unrestricted randomization within subdistrict strata. We compared the variance of differences between intervention and control groups in the stratified analysis with the unstratified analysis to estimate relative efficiency.

The subdistrict stratified analysis improved efficiency for most outcomes compared to an unadjusted analysis, but for nearly every outcome the pair matched estimator had the lowest variance (highest relative efficiency) − in many cases, the subdistrict stratified analysis was substantially less efficient compared with the pair matched analysis (Fig. 4). For five outcomes in Bangladesh, all efficiency gains were lost with subdistrict stratification. These results show that fine stratification by geography obtained through pair matching has potential to further improve efficiency compared with a design that stratifies on small administrative units, which implies there is substantial outcome variation at spatial scales below subdistrict. To further interrogate this result, we estimated spatial outcome correlation by distance using a universal kriging model with a semiparametric smooth of control cluster means (Methods). Spatial outcome correlation fell to zero by 10-20 km for nearly all outcomes (Supplementary Figs. 9, 10), demonstrating that spatial correlation was generally below subdistrict scale.

Another alternative randomization strategy is to pair match clusters using baseline measures of the primary outcome[3]. The rationale is that a baseline measure will likely be correlated with the primary outcome and would thus improve efficiency. If pair matching by a baseline covariate results in perfect balance in the covariate between groups, then the within-pair outcome correlation can be approximated by the squared correlation between the covariate and the outcome (Methods)[10]. We used this relationship to approximate the relative efficiency that could be obtained had each trial pair matched by mean length-for-age z, a primary outcome. The estimates are optimistic since

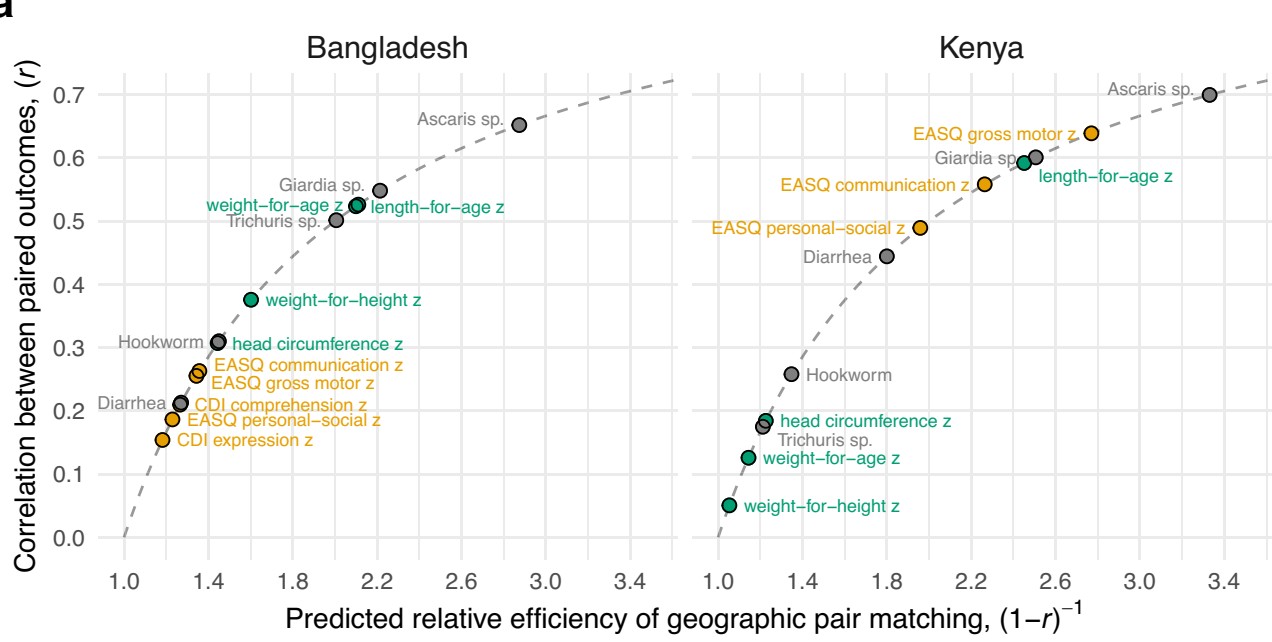

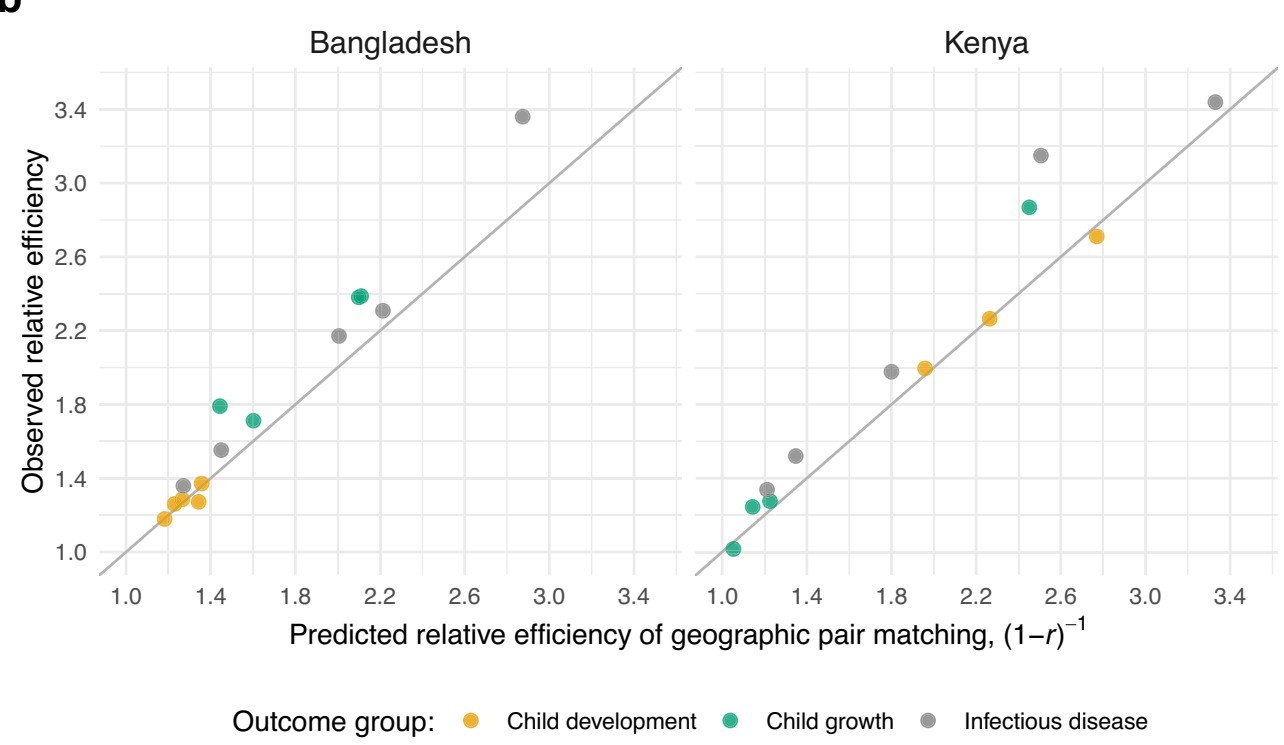

**Fig. 2 | Relative efficiency of geographic pair matching compared to an unmatched design in the Bangladesh and Kenya WASH Benefits trials. a** Paired outcome correlation across geographically matched pairs (n = 90 in Bangladesh, n = 72 in Kenya), translated into predicted relative efficiency for 14 child development, child growth, and infectious disease outcomes. Dashed lines show the $(1-r)^{-1}$ function, the predicted relationship between pair-wise correlation ($r$) and relative efficiency. **b** Observed relative efficiency of a the non-parametric, pair matched estimator versus gains predicted based on the paired outcome correlation in panel a. The observed relative efficiency used an unmatched analysis as the basis for comparison (Methods). A solid line marks the 1:1 axis. Correlation estimates based on outcomes weighted by sample sizes of each pair. MacArthur-Bates Communicative Development Inventory (CDI) comprehension and expression were only measured in the Bangladesh trial. Created with notebooks https://osf.io/pdver and https://osf.io/d2x3b.

they rely on two strong assumptions: (i) correlation between length-for-age z and the other outcomes measured contemporaneously at the end of the trial is the same as the correlation between each cluster's mean baseline length-for-age z and other outcomes measured 2 years later and (ii) perfect matching on length-for-age z.

We estimated that had the trials pair matched exactly on cluster-level mean length-for-age z, there would be large efficiency gains for weight-for-age z (highly correlated with length-for-age z) but there would have been no efficiency gains for more than half of the outcomes we studied. Compared with relative efficiency expected under a

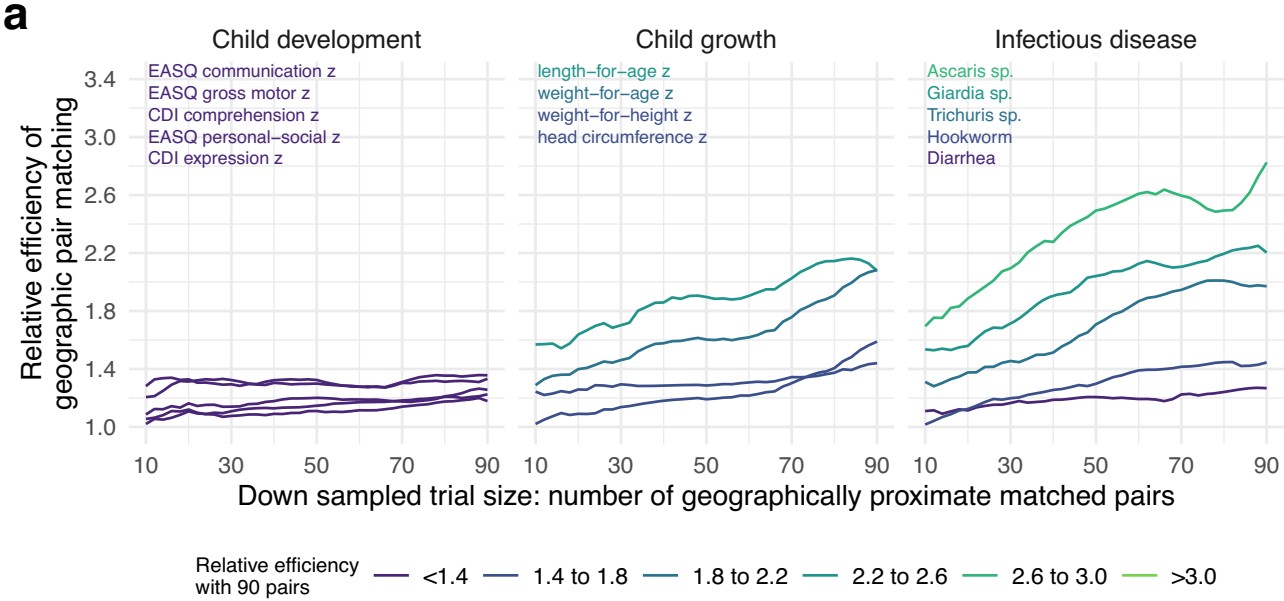

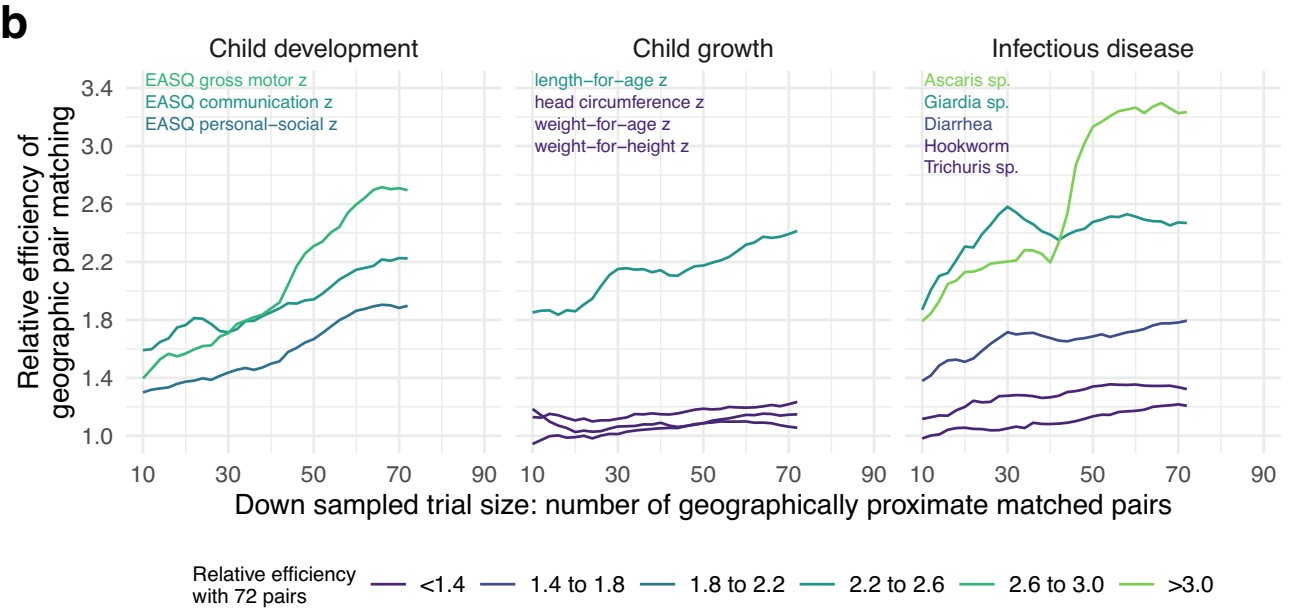

**Fig. 3 | Relative efficiency of geographic pair matching across resampled trials of varying size. a** Relative efficiency of geographic pair matching compared with an unmatched design by number of geographically proximate matched pairs in the Bangladesh trial. Lines represent mean relative efficiency over 1000 bootstrap resampled subsets of geographically proximate matched pairs in samples ranging from 10 to 90 pairs. Outcome labels in each panel are ordered and colored by relative efficiency with 90 pairs. **b** Similar estimates of mean relative efficiency over 1000 bootstrap resampled subsets of different sizes in the Kenya trial, ranging from subsamples of 10 to 72 pairs. MacArthur-Bates Communicative Development Inventory (CDI) comprehension and expression were only measured in the Bangladesh trial. Created with notebook https://osf.io/n276c.

pair matched design using length-for-age z to match, we found that geographic pair matching led to substantially higher efficiency gains for all but one outcome (Supplementary Fig. 11). This result suggests that if trials wish to optimize their design around a single outcome or a small number of highly correlated outcomes then pair matching by that outcome at baseline could be superior to geographic pair matching, but that geographic pair matching leads to far broader efficiency gains across diverse outcomes.

### Estimates of spatially varying effect heterogeneity
A unique feature of pair matched cluster randomized trials is that each pair provides an estimate, albeit noisy, of the average treatment

effect[15]. In geographic pair matched designs, pair-level effects are georeferenced by their location, enabling non-parametric or semi-parametric summaries of spatially varying heterogeneity in treatment effects that would not be possible without more elaborate, parametric modeling. To illustrate the potential utility of this type of analysis in geographically pair matched trials, we chose one illustrative outcome in each trial where the nutrition intervention led to lower levels of infection and where there was spatial heterogeneity in control group outcomes − diarrhea in Bangladesh[4] and *Ascaris* sp. infection in Kenya[26]. These were not pre-specified analyses, but instead are intended to demonstrate the methodology. We used a universal kriging approach to spatially interpolate pair-level average treatment effects.

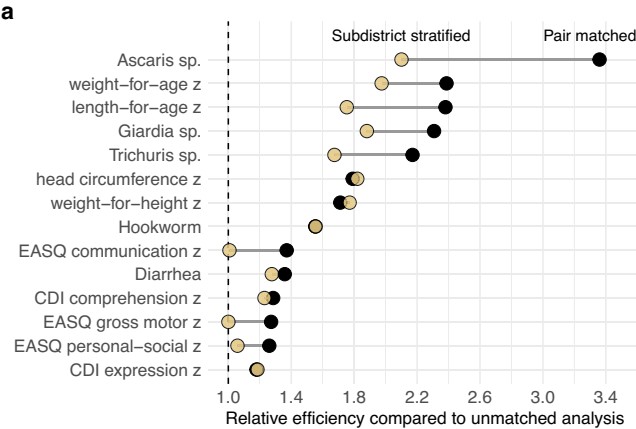

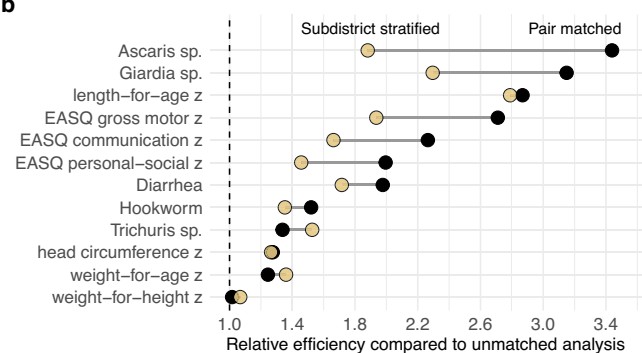

**Fig. 4 | Relative efficiency of geographic pair matching and subdistrict stratified estimators. a** Estimates from the Bangladesh trial (90 matched pairs in 19 subdistricts) for 14 outcomes, sorted by the relative efficiency of the pair matched estimator. **b** Estimates from the Kenya trial (72 matched pairs in 10 subdistricts) for 12 outcomes, sorted by the relative efficiency of the pair matched estimator. MacArthur-Bates Communicative Development Inventory (CDI) comprehension and expression were only measured in the Bangladesh trial. In both panels, relative efficiency was estimated as the ratio of the variance between a non-parametric, unmatched estimator and each alternative estimator. Created with notebooks https://osf.io/89g7m and https://osf.io/d2x3b.

The model makes no assumptions except for smoothness in the treatment effects across geography and spatial outcome correlation following a Matérn function (Methods). From the geostatistical model fit, we simulated predictions at each location to estimate an approximate posterior probability of treatment benefit throughout each study region – a measure that combines the magnitude and uncertainty of predicted treatment effects.

Diarrhea prevalence varied over the Bangladesh study region (Fig. 5a), with spatially varying average treatment effects that were larger in regions with higher prevalence (Fig. 5b). Posterior probabilities of a nutrition benefit showed highest probability in the northeastern part of the study region (Fig. 5c). *Ascaris* sp. infection in control clusters was spatially heterogeneous in the Kenya trial (Fig. 5d), but there was minimal heterogeneity in average treatment effects (Fig. 5e) or posterior probabilities of a treatment benefit (Fig. 5f) which suggests a more homogeneous effect throughout the study region.

A natural extension of the previous analyses in pair matched designs is to examine heterogeneity in pair-level average treatment effects by spatially varying covariates. In a pair matched design, pair-level conditional average treatment effects can be examined over continuously varying effect modifiers. In principle, geo-located pair-level differences can be spatially joined to any continuous surface such as environmental covariates measured through remote sensing or spatially modeled sociodemographic layers. Population remoteness from urban areas is one such potential effect modifier, since

remoteness could influence both exposure to environmentally mediated pathogens and access to treatment once infected. We joined pair-level estimates to 1 km gridded surfaces of modeled travel time to cities in 2015[28] – in Bangladesh, we estimated travel time to Dhaka using the underlying friction surface[29] because Bangladesh's dense settlement pattern meant there was almost no variation in the previously generated travel time to cities layer. We used the same outcomes as the previous section to illustrate the approach – diarrhea in Bangladesh and *Ascaris* sp. in Kenya. Once pair-level treatment effects were spatially joined to continuous measures of population remoteness, we used locally weighted regression to examine the relationship between remoteness and the effects of the nutrition intervention in each trial.

There was substantial variation in pair-level differences in Bangladesh, but non-parametric, locally weighted regression fits show higher diarrhea prevalence and larger reductions in the nutrition group in pairs further from Dhaka (Fig. 6). In Kenya, *Ascaris* sp. prevalence was lower in more remote clusters, and average treatment effects were slightly larger among pairs at intermediate distances of 10-20 minutes from the nearest city, albeit substantial variation in pair-level differences (Supplementary Fig. 12).

## Discussion

Across a broad set of child health outcomes measured in large-scale cluster randomized trials in Bangladesh and Kenya, we found that geographic pair matching led to substantial improvements in statistical efficiency. Of 26 outcomes assessed across the two trials, all had positive efficiency gains and 11 had relative efficiencies between 2.0 and 3.3, meaning an unmatched trial would have needed to enroll at least 2 to 3 times as many clusters to achieve the same level of precision as the pair matched design. In large-scale biomedical field trials, such differences in sample size are non-trivial. In the Bangladesh trial, for example, that would have meant 180 to 270 additional clusters (1440 to 2160 newborns) per group, a prohibitive increase in scale based on logistics and costs. Additionally, we demonstrated how geographic pair matched randomization enabled unique, semi-parametric analyses of spatial effect heterogeneity not possible through an unmatched design. Effect heterogeneity is typically relegated to exploratory, secondary analyses of cluster randomized trials. Yet, geographically explicit treatment effect estimates, even if exploratory, can help inform subsequent scale-up of programs that often follow large-scale trials of interventions that prove effective, as has recently been recommended for the nutritional intervention studied here[30].

Our empirical results confirm the theoretical predictions for efficiency gains through pair matched randomization and provide insights into the unique benefits of matching by geography in large-scale trials. The Bangladesh and Kenya trials' pair matched design was inspired by theoretical and empirical arguments developed in the context of public policy and health economics, which showed large benefits of pair matching (not by geography) in Mexico's Seguro Popular universal health insurance trial[14–16]. Near-universal increases in statistical efficiency demonstrated herein show substantial gains from geographic pair matching for child health outcomes where all but one pair-wise correlations exceeded the "break-even" correlation for a design with as few as 20 clusters, or 10 matched pairs (Supplementary Fig. 4a)[10]. Unique to geographic pair matching, we showed that efficiency gains scale with a trial's geographic footprint, with higher levels of outcome clustering (ICC, Moran's I), and with higher outcome prevalence (binary outcomes). These results suggest that geographic pair matching capitalizes on spatial variation in underlying sociodemographic and environmental conditions that ultimately influence child growth, development, and infectious disease transmission. The consistency of results across diverse outcomes from two trials in different populations suggests generalizability of the empirical findings.

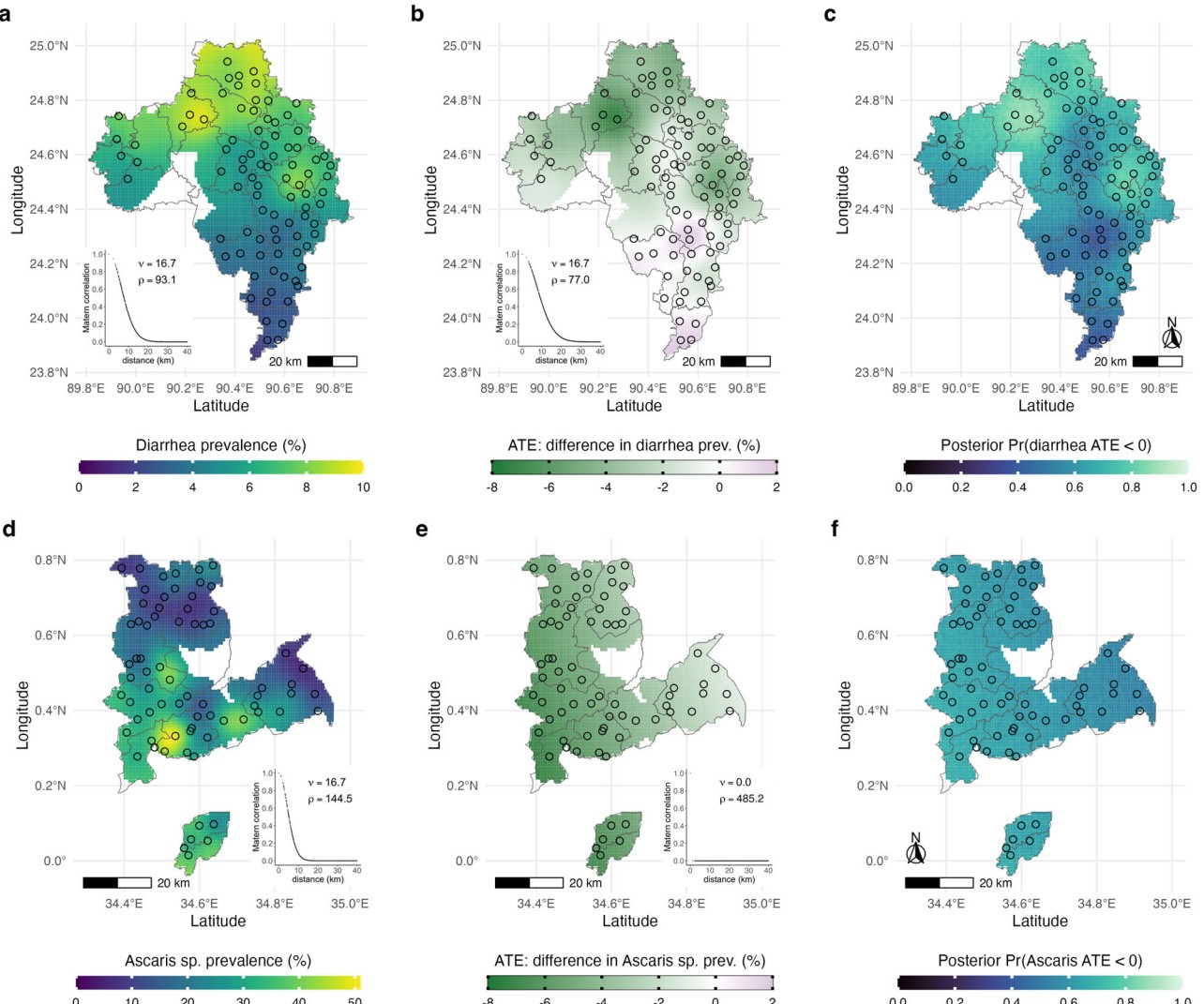

**Fig. 5 | Spatial heterogeneity of intervention effects in geographically pair matched trials. a** Spatially heterogeneity in diarrhea prevalence in the control group in the WASH Benefits Bangladesh trial, visualized through universal outcome kriging with a Matérn spatial correlation structure. **b** Spatially smoothed average treatment effects (ATE) of matched-pair differences of diarrhea prevalence comparing nutrition and control clusters in the Bangladesh trial. **c** Posterior probability that the nutrition intervention reduced diarrhea in Bangladesh, derived from the geostatistical model used to smooth the ATE in panel b. **d** Spatial heterogeneity in *Ascaris* sp. infection prevalence in the Kenya trial. **e** Spatially smoothed ATE of matched-pair differences of *Ascaris* sp. prevalence comparing nutrition and control clusters in the WASH Benefits Kenya trial. **f** Posterior probability that the nutrition

intervention reduced *Ascaris* sp. infection in Kenya, derived from the geostatistical model used to smooth the ATE in panel e. Smoothed surfaces at 1 km resolution were estimated using a geostatistical model with Matérn spatial covariance, trimmed by study subdistrict boundaries and a 10 km buffer around matched pair centroids. Insets of panels a, b, d and e show estimated parameters and Matérn correlation function with distance between matched pairs, illustrating no spatial correlation in the ATE for *Ascaris* sp. in Kenya. Points represent matched pair centroids and lines demark subdistricts in the study regions (zillas in Bangladesh, sub-counties in Kenya). In panels c and f, posterior probabilities were estimated from 1,000 simulation replicates at each location, drawn from the geostatistical model fits of the ATE (Methods) Created with notebook https://osf.io/j9r4k.

Although we present results from only two settings, the observed, spatial clustering of health determinants and outcomes in across myriad populations[19] suggests that large epidemiologic field trials with spatially heterogeneous outcomes would likely benefit from geographic pair matching in the design.

Logistical considerations are magnified in large-scale trials, and any added complexity in a trial's design can make its logistics more difficult, threatening its internal validity. Pair matching using geographic location has the key logistical advantage that it is immediately available and requires no baseline measurements in the study population. An additional benefit of geographic pair matching is that geography alone captured variation across diverse child health outcomes in the contexts we studied. For complex interventions, such as the nutritional and environmental improvements delivered in the WASH Benefits trials, there are often many outcomes of interest beyond the

trial's primary endpoint. Geographic pair matching leads to much broader and, in general, larger efficiency gains across outcomes compared to gains expected under pair matching by length-for-age z, a primary outcome (Supplementary Fig. 11). Moreover, we found that pair matched randomization using each cluster's precise location led to far larger efficiency gains for most outcomes than if the trials used subdistrict stratification, which suggests stratified randomization, even within small administrative areas, leaves substantial power on the table (Fig. 4).

Geographic pair matching was universally beneficial with respect to efficiency gains in the outcomes we studied, but the magnitude of efficiency gains is unlikely to be known at the design stage of the trial. These analyses suggest that efficiency gains are more likely in larger scale trials, and if information is known about outcome spatial heterogeneity (ICC, Moran's I) then outcomes with more clustering are

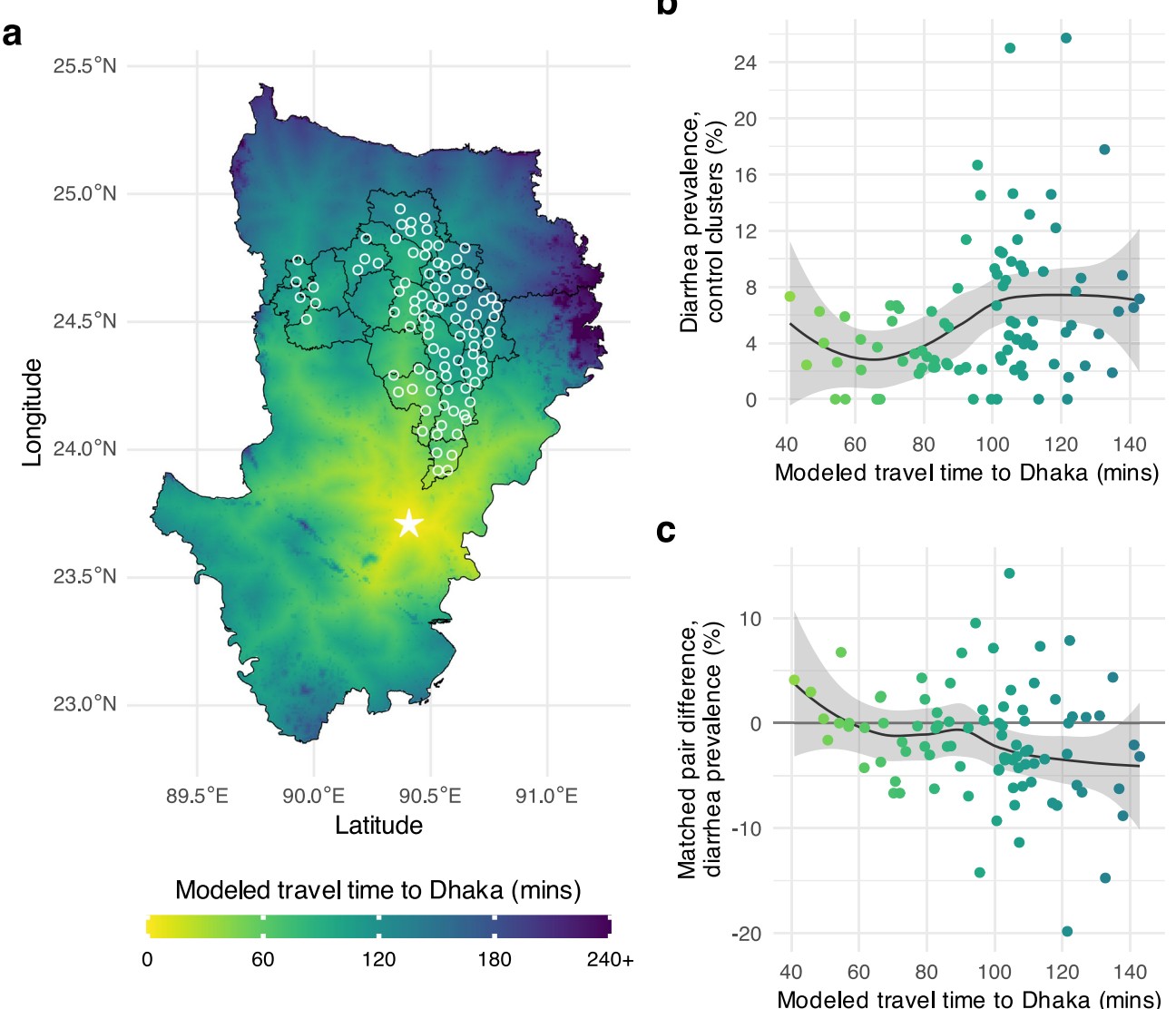

**Fig. 6 | Heterogeneity in the effect of nutrition on diarrhea prevalence by travel time from Dhaka, Bangladesh. a** Modeled travel time in minutes at 1 km² resolution between Dhaka (marked by a star) and the 90 WASH Benefits Bangladesh matched pair centroids (white circles). Black lines mark subdistricts (zillas). **b** Diarrhea prevalence in control clusters by travel time to Dhaka. The line represents a non-parametric locally weighted regression fit, and the shaded band its approximate pointwise 95% confidence interval **c** Matched pair differences in diarrhea prevalence (nutrition − control) by travel time to Dhaka. The line represents a non-parametric locally weighted regression fit, and the shaded band its approximate pointwise 95% confidence interval. In panels b and c, points are colored by the surface in panel a. Created with notebook https://osf.io/fmgex.

more likely to benefit from geographic pair matching. Alternatively, trials that use geographic pair matching could ignore potential efficiency gains at the design stage for primary outcomes and use it as an insurance strategy for bias and power. The broader benefits beyond primary outcomes may be useful given that trials are rarely independently powered for secondary outcomes. Indeed, both WASH Benefits trials used geographic pair matching in the randomization, but their sample size calculations for primary outcomes ignored pair matching to be conservative, given this uncertainty[21].

Semi-parametric analyses of spatial effect heterogeneity illustrated how geo-located, pair-level estimates of the ATE, albeit noisy on their own, can be smoothed to create a continuous treatment effect surface and can be joined to any spatially continuous covariate to assess effect heterogeneity using a simple, locally weighted regression. In principle, such analyses at fine spatial scales could be done without geographic pair matching but would require fitting separate outcome models for each group (intervention, control) and then taking the difference between model fits to estimate the ATE − an approach that would have more variability because of the joint uncertainty in the separate outcome processes. An intermediate strategy that averages cluster-level outcomes within well-defined geographic units (such as subdistricts) would be possible under simple randomization but would lose some of the very fine scale information that a continuously smoothed ATE surface provides if there is variation within administrative areas, as there was in the case of diarrhea in Bangladesh (Fig. 5b). Beyond population remoteness, we did not study mechanisms that underlie the observed spatial effect heterogeneity in Bangladesh. Since geography integrates many environmental and socioeconomic characteristics, the mechanism likely multifactorial but such analytic extensions could provide useful information for future intervention planning.

Our results identified situations where geographic pair matching might be less beneficial. Paired outcome correlations estimated in these trials would not favor geographic pair matching with fewer than

10 pairs. Pair matching did not appreciably improve efficiency for binary outcomes with prevalence below 5%, likely due to minimal variation in the outcome (Supplementary Fig. 5c). Future studies could assess whether geographic pair matching improves efficiency for rare outcomes in trials with larger cluster sizes or more longitudinal measurements within clusters, effectively providing more information to estimate cluster-level means.

Future studies could also assess optimal distance between matched pairs, which should depend on spatial outcome correlation patterns and the potential for spillover between clusters. Spatial outcome correlation disappeared by 10–20 km for almost all outcomes studied here (Supplementary Figs. 9 and 10), suggesting that geographically close pairing (~1 km) was beneficial for efficiency gains. Matched pairs separated by larger geographic distances might benefit less from pair matching depending on the scale of geographic variation in the outcome. As with unmatched cluster randomized trials, a key assumption of matched pair trials is no interference (spillover) between matched pairs[15]. Although clusters were relatively close in both trials, they were separated by ≥1 km to prevent contamination between them and empirical estimates in the original trials demonstrated an absence of spillover effects on infectious disease, child growth, measures of environmental contamination and behavioral outcomes[4,5,31].

There are some caveats to these analyses. Cluster sizes were small in both trials and within-pair sample sizes were generally well balanced between groups by design. It is therefore unclear how well our results would generalize to trials with larger clusters that have hundreds or thousands of measurements or to trials with large between-group differences in paired cluster sizes. Ensuring similar cluster sizes within pairs has theoretical advantages in terms of bias in the non-parametric estimator[15], and validity of permutation tests in matched-pair designs[32] so should be preferred in matched designs whenever feasible. Similar cluster sizes within pairs also leads to a balanced number of individuals in each group, which is more statistically efficient compared to an imbalanced design. Another caveat is that all efficiency comparisons focused on non-parametric, cluster-level estimators of the ATE. Regression-based estimators that use individual-level responses with appropriate variance corrections provide an alternative analysis approach[3,15], but since we used weighted estimators we would not expect appreciable differences in variance estimates. The trials did not have baseline outcome measurements as children were not yet born so we could not assess whether additional gains might be possible by combining geographic pair matched (or stratified) randomization with additional regression adjustment for baseline outcomes. Additional gains may be possible by controlling for residual outcome variation not captured in the matched or stratified randomization, but care must be taken in selection of adjustment covariates and the analysis methods add complexity compared with the unadjusted estimators used here[33]. Additionally, we did not study the effect of geographic pair matching on efficiency of permutation tests[34,35]. Inference based on the paired t test and the permutation test is very similar for large sample sizes anticipated in large-scale cluster randomized trials, which we demonstrated for primary outcomes in both trials[4,5], but it may be an interesting topic for future studies. Finally, the enrollment and pair matching of geographically proximate clusters occurred sequentially over the course of a year, meaning that clusters were effectively pair matched on geography and calendar time. This means that observed spatial heterogeneity could also embed seasonal heterogeneity for seasonally-varying outcomes, such as diarrhea in Bangladesh. Trials that wish to separately study spatial heterogeneity and temporal heterogeneity would need to measure geographically pair matched clusters in a more random order, rather than sequentially over a study region.

When considering the generalizability of the findings, note that the results arise from two community-based trials with a relatively large number of pairs, small cluster sizes, and outcomes that were specific to the rural Bangladesh and Kenya contexts. The results should not be extrapolated to trials with fewer than 10 clusters where the relative efficiency could be worse if geographic pair matching induces a weak correlation in outcomes (r < 0.11). Moreover, it is unclear whether geographic pair matching of randomized units other than communities, such as healthcare centers, would obtain similar gains in efficiency or could be used to study fine-scale spatial heterogeneity in treatment effects as we have done here. Finally, the relationship between efficiency gains and measures of intra-cluster correlation and spatial autocorrelation could be different in trials with larger clusters — a recent simulation study showed that increasing cluster size may play a role achieving effective matching even when the ICC approaches zero[36].

In conclusion, this empirical analysis of geographic pair matching in large-scale trials of in rural Bangladesh and Kenya confirmed predictions based on statistical theory and demonstrated substantial, universal gains in statistical efficiency across child health outcomes that were often the equivalent of increasing the sample size by 2-3 times compared with an unmatched design. Geographically pair matched designs enabled estimation of continuous effect surfaces and effect heterogeneity by continuously varying spatial covariates with almost no modeling assumptions — a valuable extension in large-scale trials where inference at smaller geographic scales may be warranted. Based on these substantial benefits with few logistical or analytical costs, geographic pair matching should be considered in the design of future large-scale, cluster randomized trials.

## Methods
### Inclusion and ethics
Trial protocols were reviewed and approved by ethical review committees at the International Centre for Diarrhoeal Disease Research, Bangladesh (PR-11063), the Kenya Medical Research Institute (protocol SSC-2271), University of California, Berkeley (protocols 2011-09-3652, 2011-09-3654), and Stanford University (protocols 23310, 25863). All participants provided informed consent. Investigators from Bangladesh and Kenya led the trials in partnership with academic partners in the United States and have continued to provide substantive input into the design and interpretation of secondary analyses, including the present study.

### Study design
The WASH Benefits study protocol and primary outcome papers include details of the design, rationale, and intervention procedures[4,5,21]. The Bangladesh and Kenya trials shared overall objectives and major design features, but were designed as separately powered, replicated trials. In Bangladesh, field teams identified groups of 8 pregnant mothers in their second trimester living geographically close enough for a local health promoter to visit them regularly. This formed a cluster, and randomization was at the cluster level to enable a single health promoter to deliver a consistent intervention to all 8 pregnant mothers and their children. The field team traveled at least 1 km before starting a new cluster to prevent between-cluster spillover effects[31]. Geographically proximate blocks of 8 clusters were pair matched and randomized to a double-sized control group (2 clusters) or one of 6 intervention groups, described below. The Bangladesh trial included 90 blocks of 8 clusters, for a total of 720 clusters (Fig. 1a).

In Kenya, the design was almost identical to Bangladesh, but clusters were slightly larger (12 pregnant mothers per cluster on average) and geographically pair matched blocks included 9 clusters rather than 8 to allow for a passive control group. In Kenya, the double-sized control group included monthly mid-upper arm circumference measurements and the passive control group included no visits to assess whether visits alone influenced outcomes (there was no difference)[5]. The Kenya trial included 89 blocks of up to 9 geographically pair matched clusters, but 21 blocks were incomplete so the trial enrolled a total of 702 clusters (Fig. 1a).

The six intervention groups included: household water treatment (W), improved latrines for all households in intervention compounds (S), improved handwashing facilities with soap at the latrine and kitchen (H), improved nutrition for birth cohort children (N), combined WSH and combined WSH + N. The improved nutrition intervention included promotion of Infant and Young Child Feeding recommendations such as exclusive breastfeeding through age 6 months and continued breastfeeding through age 24 months, plus a daily Small Quantity Lipid Based Nutrient Supplement (SQ-LNS) from ages 6 to 24 months. Interventions primarily focused on children in the birth cohort and their household. Households were located within multiple-household family compounds and the sanitation intervention improved latrines for the entire compound. For all interventions, community health promotors who lived in study communities delivered an in-depth behavior change program that was developed over several years of pilot studies ahead of the main trials.

Since our focus in this analysis was on design methodology and statistical properties of geographically pair matched designs, we restricted the analysis to clusters in the double-sized control group and those that received the nutrition intervention (N, WSH + N). This created a balanced design within matched pairs, with each geographic pair including 4 clusters (Fig. 1c). All 90 blocks in the Bangladesh trial were complete but in the Kenya trial we limited the analysis to 72 pair matched blocks that were complete with respect to the 4 clusters of interest. Clusters that received the nutrition intervention were analyzed as a single intervention group as there has been no evidence for differences between the N and WSH + N group for any outcome we included, with outcome-specific references provided in the following section. Child sex was not considered in the present analyses as all age-eligible children were enrolled and analyzed in the trials, but sex has been provided in individual level Source Data files.

## Outcome measurements

The analysis included 14 previously published endpoints in the trials (Supplementary Tables 1 and 2). We chose outcomes that were measured in the full birth cohort (not a substudy), included continuous and binary measures, and represented a broad range of child health outcomes measured in biomedical field trials (growth, development, infectious diseases). The number of measurements varied by outcome due to slightly different measurement strategies for each.

Child length, weight, and head circumference measurements were included from each trial's final visit when children were approximately 24 months old. Anthropometric measures were converted into length-for-age z, weight-for-age z, weight-for-height z, and head circumference z using the median of triplicate measurements, child's age in days, and the 2006 World Health Organization reference standards[4,5,37].

The trials used the Extended Ages and Stages Questionnaire (EASQ) to measure gross motor, communication, and personal-social skills when children were approximately 24 months old. EASQ scores in the control group were normalized (mean 0, standard deviation 1) to create z scores within 2-month age bands for the study population[22,23]. The Bangladesh trial additionally measured language development using a locally validated version of the MacArthur Bates Communicative Development Inventory (CDI) for verbal comprehension and expression[22].

Both trials measured diarrhea using caregiver reported symptoms when children were approximately 12 and 24 months old. Symptoms were measured using a 7 day recall period and diarrhea was defined as three or more loose or watery stools in 24 h or a single stool with blood[4,5]. The trials measured parasite infection in whole stool among birth cohort children when they were approximately 24 months old plus older children living in study compounds: in Kenya up to one older child (ages 3–15 years) and in Bangladesh up to two older

children (ages 3–12 years), with preference given to older siblings living in the birth cohort child's household[25,26]. Stool specimens were tested for soil transmitted helminth infections including: *Ascaris lumbricoides, Trichuris trichiura*, and hookworm (*Ancylostoma duodenale, Necator americanus*) using the double-slide Kato-Katz technique. Stool specimens were additionally tested for *Giardia duodenalis* infection using qPCR in Bangladesh and ELISA in Kenya[24,26].

## Basic assumptions

Formal statistical assumptions of pair matched cluster randomized trials have been defined[15]. In brief, we assume that treatment is randomized at the cluster level and that potential outcomes of individuals within clusters do not depend on the treatment status of other clusters, including matched pair clusters. The second assumption (no contamination or spillover between clusters) is reasonable in this context as clusters were separated by ≥1 km to prevent spillover, and an absence of between-cluster spillover has been previously demonstrated for several outcomes in the trials[4,5,31]. Finally, we assume that clusters represent a random sample from the larger population but all individuals within a cluster are observed. This reflects the trials' design since clusters consisted of all eligible mothers and their newborn children and has been referred to as the unit average treatment effect[15].

## Relative efficiency as a function of pair-wise outcome correlation

We defined relative efficiency as the ratio of the variance of group differences in a design with unrestricted randomization versus a design with matched pair randomization. For outcome $Y_0$ measured in the control group with variance $\sigma_0^2$ and $Y_1$ measured in the intervention group with variance $\sigma_1^2$, in an unmatched design $Y_0$ and $Y_1$ are independent and so the variance of the difference is

$$\sigma_U^2 = \sigma_1^2 + \sigma_0^2 \tag{1}$$

Pair matching can induce correlation between outcomes, such that the variance of the difference is then

$$\sigma_P^2 = \sigma_1^2 + \sigma_0^2 - 2Cov(Y_1, Y_0). \tag{2}$$

Defining the pair-wise correlation as, $r = Cov(Y_1, Y_0)/(\sigma_1 \sigma_0)$, the variance of matched pair differences is

$$\sigma_P^2 = \sigma_1^2 + \sigma_0^2 - 2\sigma_1 \sigma_0 \cdot r. \tag{3}$$

If the variance is equal in the two groups, the unmatched variance is $2\sigma^2$ and the matched pair variance is $2\sigma^2(1-r)$, with relative efficiency:

$$Reff = \sigma_U^2/\sigma_P^2 = (1-r)^{-1}. \tag{4}$$

If variances are unequal in the two groups, $Reff = (1-x)^{-1}$ where $x = 2Cov(Y_1, Y_0)/(\sigma_1^2 + \sigma_0^2)$. Thus, relative efficiency scales with the correlation between the paired outcomes. Note that this simple relationship holds under a model for the paired $t$ test[27], and for nonparametric estimators of pair matched differences[15]. Furthermore, $r$ can represent the unweighted correlation or weighted correlation using, as in this study, the number of individuals measured in each pair as weights. Our primary analyses favored a weighted correlation since it aligned much more closely with empirical efficiency gains (Fig. 2b, Supplementary Fig. 4b).

To see the relationship between relative efficiency and trial sample size, let $S$ be the sample standard deviation of cluster-level outcomes with $m_U$ clusters enrolled in an unmatched design ($\sigma_U^2 = S^2/m_U$) and $m_P$ clusters enrolled in a pair matched design ($\sigma_P^2 = S^2/m_P$).

Assume equal variance in the intervention and control group and set the variances of the unmatched and pair matched designs equal: $2\sigma_U^2 = 2\sigma_P^2(1 - r)$, $2\frac{S^2}{m_U} = 2\frac{S^2}{m_P}(1 - r)$, and $m_U = m_P(1 - r)^{-1}$. The number of clusters required in an unmatched design, $m_U$, is thus the number required in a pair matched design $m_P$ multiplied by the relative efficiency $Reff = (1 - r)^{-1}$.

We estimated group means for control and nutritional intervention groups within matched pairs (90 in Bangladesh, 72 in Kenya) and then estimated a weighted correlation between outcomes with weights equal to the total number of children in the matched pair. To estimate 95% confidence intervals for weighted correlation of within-pair outcomes and corresponding relative efficiency, we used a non-parametric bootstrap resampling approach, resampling pairs with replacement and 1000 iterations.

For labeling in figures, we also computed the "break-even" correlation for different trial sizes−the minimum correlation in paired outcomes matching must induce for the matched pair design to be able to detect smaller differences compared with an unmatched design[10]. We assumed 80% power and 5% Type I error in the calculation. For example, the break-even correlation for 10 pairs is $r = 0.11$.

## Relative efficiency estimates from pair matching on other variables

Many trials might consider pair matching on a baseline covariate $x$, such as a baseline measure of the primary outcome, to ensure balance and gain efficiency in the primary analysis. If the primary outcome is denoted $y$, then under a perfect match between clusters on $x$, the within-pair outcome correlation is equal to the squared correlation between $x$ and $y$[10]: $r_{yy} = (r_{xy})^2$. We estimated the correlation in pair level means in control clusters for all outcomes included in the analysis.

To provide an estimate of the efficiency gains possible under pair matching by the trials' primary outcome, length-for-age z, we translated the correlation between length-for-age z score ($x$) and each outcome ($y$) to an estimate of the within-pair outcome correlation using the above relationship and then to relative efficiency compared with an unmatched design: $(1 - r_{yy})^{-1}$. The estimates of relative efficiency are approximate and optimistic because they make two strong assumptions: perfect matching in cluster-level means of length-for-age z, and the correlation between length-for-age z and other outcomes measured contemporaneously is the same as if we had access to true baseline means in length-for-age z, measured two to three years in the past.

## Relative efficiency of pair matched and subdistrict stratified estimators

We compared empirical variances of the ATE estimated in the trials using different estimators: pair matched analysis, an unmatched analysis, and a subdistrict stratified analysis. Our parameter of interest was the mean difference in outcomes between children that received the nutritional intervention, $Y_1$, versus those that did not, $Y_0$: $\psi = E[Y_1 - Y_0]$. Non-parametric estimators and their variance for pair matched and unmatched designs follow those proposed by Imai and colleagues[15]. Given a pair matched design, ignoring the pair matching in the analysis provides an unbiased estimate of the variance under an unmatched design[15].

For $m$ matched pairs, $c_k$ clusters in matched pair k, and $n_{jk}$ children in cluster $j$ and pair $k$, the sample ATE is:

$$\psi_{ATE}^M = \frac{1}{n}\sum_{k=1}^{m}\sum_{j=1}^{c_k}\sum_{i=1}^{n_{jk}}(Y_{1ijk} - Y_{0ijk}). \tag{5}$$

A non-parametric, unbiased estimator is the weighted average of pair-level differences in group means. The treatment assignment of cluster $j$ in matched pair $k$ is $A_{jk}$ (1 intervention, 0 control):

$$\widehat{\psi}_{ATE}^M = \frac{1}{\sum_{k=1}^{m}w_k} \cdot \sum_{k=1}^{m}w_k\left[\left(\frac{\sum_{j=1}^{c_k}n_{jk}A_{jk}\sum_{i=1}^{n_{jk}}Y_{ijk}}{\sum_{j=1}^{c_k}n_{jk}A_{jk}}\right) - \left(\frac{\sum_{j=1}^{c_k}n_{jk}(1 - A_{jk})\sum_{i=1}^{n_{jk}}Y_{ijk}}{\sum_{j=1}^{c_k}n_{jk}(1 - A_{jk})}\right)\right] \tag{6}$$

with weights defined as the number of children in the matched pair, $w_k = \sum_{j=1}^{c_k}n_{jk}$. A conservative estimate of the variance is:

$$Var(\psi_{ATE}^M) = \frac{m}{(m-1)n^2}$$
$$\cdot \sum_{k=1}^{m}\left\{w_k\left[\left(\frac{\sum_{j=1}^{c_k}n_{jk}A_{jk}\sum_{i=1}^{n_{jk}}Y_{ijk}}{\sum_{j=1}^{c_k}n_{jk}A_{jk}}\right) - \left(\frac{\sum_{j=1}^{c_k}n_{jk}(1 - A_{jk})\sum_{i=1}^{n_{jk}}Y_{ijk}}{\sum_{j=1}^{c_k}n_{jk}(1 - A_{jk})}\right)\right] - \frac{n\widehat{\psi}_{ATE}^M}{m}\right\}^2 \tag{7}$$

In a study with a total of $n$ children and $c$ clusters, and $n_j$ children in cluster $j$, we defined the unmatched ATE for potential outcomes of child $i$ in cluster $j$ as:

$$\psi_{ATE}^U = \frac{1}{n}\sum_{j=1}^{c}\sum_{i=1}^{n_j}\left(Y_{1ij} - Y_{0ij}\right). \tag{8}$$

The non-parametric estimator is:

$$\widehat{\psi}_{ATE}^U = \frac{1}{n}\sum_{j=1}^{c}\sum_{i=1}^{n_j}\left[A_j Y_{ij} - \left(1 - A_j\right)Y_{ij}\right] \tag{9}$$

A conservative estimate of the variance is a function of variability in weighted cluster level means:

$$Var(\psi_{ATE}^U) = \frac{2c}{n^2}\left[Var\left(\widetilde{w}_j \cdot \overline{Y_{j1}}\right) + Var\left(\widetilde{w}_j \cdot \overline{Y_{j0}}\right)\right], \tag{10}$$

where $\overline{Y_{ja}} = \sum_{i=1}^{n_j}Y_{ija}/n_j$ for $a = 0,1$. We used weights equal to the cluster size, $\widetilde{w}_j = n_j$.

As a final comparator, we considered a subdistrict stratified ATE, defined as

$$\psi_{ATE}^S = \sum_{s}[E(Y|A=1, S=s) - E(Y|A=0, S=s)] \cdot Pr(S=s) \tag{11}$$

for geographic subdistricts $S$ that included 19 zillas in Bangladesh (administrative level 3) and 10 sub-counties in Kenya (administrative level 2). These are the smallest administrative levels recognized in each country above the village level. We estimated this parameter using weighted least squares regression of cluster level means on a treatment indicator and subdistrict level indicators, with weights equal to the number of children in each cluster. We used the variance on the treatment coefficient in the model for estimates of relative efficiency. As an internal consistency check, we confirmed that a similar model, stratified on matched pair rather than subdistrict (equivalent to a weighted, paired $t$ test), provided identical point estimates and highly similar variance to the nonparametric, pair matched estimator.

For each outcome, we defined the observed relative efficiency as the ratio of the variance estimated in the unmatched analysis to the variance of the pair matched estimator or the subdistrict-adjusted estimator. We then compared the observed relative efficiencies with one-another, and with the relative efficiency predicted by weighted correlation estimated using pair matched outcomes, as defined in the previous section.

### Influence of trial size on efficiency gains through geographic pair matching

To study the effect of trial size on relative efficiency gains through geographic pair matching, we conducted a resampling analysis that used a modified bootstrap to down sample the trial, using smaller, geographically contiguous subsamples to represent smaller trials. We identified subsamples of size $s = 10, 12, 14, \ldots, m$ pairs in increments of 2 ($m = 90$ in Bangladesh, $m = 72$ in Kenya). Clusters were kept at their observed sizes, and so the different subsampled trials varied in size according to the number of matched pairs. For subsample of size $s$, we randomly sampled with replacement a matched pair and identified the $s - 1$ geographically closest pairs. We estimated the weighted correlation of pair matched outcomes for each randomly selected, geographically contiguous subsample, and the corresponding relative efficiency using the relationship defined above, $Reff = (1 - r)^{-1}$. We repeated the modified bootstrap resampling process 1000 times, and summarized the mean relative efficiency and the maximum distance between pairs (in km) across bootstrap replicates for each outcome and subsample of size $s$.

### Estimation of outcome intra-cluster correlation and spatial clustering

We examined outcome intra-cluster correlations (ICC), defined as ratio of between-cluster outcome variance to the total variance. A higher ICC indicates higher outcome correlation within clusters. We fit an intercept-only linear regression model on individual-level outcomes that included a random effect for cluster. We estimated the ICC as the between-cluster variance divided the total variance, and estimated its 95% confidence interval using a non-parametric bootstrap with 1000 iterations[38]. We characterized global spatial clustering in outcomes using Moran's I as a measure of spatial autocorrelation, using the inverse-distance between pair centroids as weights[39,40]. Estimates of the ICC and Moran's I were restricted to control clusters only to avoid the potential influence of intervention on each measure.

### Estimation of spatially varying effect heterogeneity

We used universal kriging, a semi-parametric smoothing technique, to spatially smooth pair-level effect estimates for one outcome in each trial: diarrhea in Bangladesh and *Ascaris* sp. infection in Kenya. We chose these two outcomes from the large list studied to demonstrate the technique because both outcomes exhibited spatial heterogeneity in underlying prevalence and were reduced by the nutrition intervention in the trials.

The kriging model conditioned on the observed pair-level treatment effects and allowed them to vary by longitude ($x$) and latitude ($y$):

$$E\left[\widehat{\psi}_{ATE}^{M} \mid x, y\right] = \alpha + \beta_1 x + \beta_2 y + S(x, y). \tag{12}$$

The $S(x,y)$ term allowed for spatial correlation in the outcomes using a Gaussian process with Matérn covariance function with smoothness $v$ and scale $\rho$ parameters. We fit the model with maximum likelihood[41]. The only assumptions of the approach are that the outcome surface is smooth and the spatial outcome correlation follows the Matérn function—thus, inference relies principally on the pair matched randomization and not on the statistical model. To visualize spatial heterogeneity in the underlying outcomes we fit the same universal kriging model to pair-level control group means for all outcomes included in the analysis.

Conditional on the geostatistical model parameter fit, we simulated 1000 predicted treatment effects at each location over a fine grid of the study area[41], within 10 km of matched pair centroids. At each location, we used the proportion of 1000 simulated outcomes where there was a reduction in prevalence due to intervention as an estimate of the posterior probability of benefit $\Pr(\psi_{ATE} < 0 \mid x, y)$ — a quantity that combines the magnitude of effect and conveys uncertainty in the predictions[42].

### Estimation of effect heterogeneity by travel time to cities

To demonstrate how geographic pair matching allows for non-parametric assessment of effect heterogeneity by continuous, spatially varying covariates, we extended the effect heterogeneity analyses of diarrhea in Bangladesh and *Ascaris* sp. infection in Kenya. We considered travel time to cities as an example effect modifier because of its potential influence on access to treatment (antibiotics, anthelmintics) or exposure to environmentally mediated pathogen transmission.

We joined pair-level differences to modeled surfaces of travel time to cities in 2015 at 1 km² grid cell resolution[28]. In Bangladesh, there was almost no heterogeneity in modeled travel time to cities given rural Bangladesh's dense settlement pattern so we estimated travel time to Dhaka using the underlying friction surface and a previously published algorithm, also at 1 km² resolution[29]. We joined the spatial layers to mean prevalence in the control clusters and pair-level differences, and then summarized the relationship between continuous travel time and mean outcomes, along with approximate ($t$ distribution) pointwise standard errors, using a locally weighted regression with default parameters in R[43].

### Reporting summary

Further information on research design is available in the Nature Portfolio Reporting Summary linked to this article.

## Data availability

De-identified data and replication files required to conduct the analyses are available through the Open Science Framework (https://osf.io/cxb5e)[44]. Geographic location data from the trials required to make maps and conduct spatial analyses are not publicly available to protect participant confidentiality. Access to geolocation data may be possible upon reasonable request to the corresponding author (ben.arnold@ucsf.edu) within ten years of publication, pending appropriate human subjects review and approval for the use of de-identified participant data (timeframe for initial response: weeks). Subdistrict boundaries used in maps and analyses were created by GADM (version 3.6, available for unrestricted use at www.gadm.org). The global friction layer and travel time to cities layer were created by the Malaria Atlas Project and accessed through the malariaAtlas R package[28,45]. The de-identified individual level data generated in this study are provided in the Source Data file. Source data are provided in this paper.

## Code availability

Computer code used to replicate the analyses is available through the Open Science Framework (https://osf.io/cxb5e)[44]. Analyses used R statistical software (version 4.3.2, 2023-10-31).

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

## Acknowledgements

This work was supported by the National Institute of Allergy and Infectious Diseases (R01AI166671 to BFA) (K01AI141616 to JBC) and the Bill & Melinda Gates Foundation (OPPGD759 to JMC). Jade Benjamin-Chung and Amy Pickering are Chan Zuckerberg Biohub investigators.

## Author contributions

Following CRediT taxonomy, conceptualization (BFA), data curation (BFA, AM, AE, CDA, AJP, JBC), formal analysis (BFA, FR, CT), funding acquisition (BFA, SPL, JMC, JBC), investigation (BFA, FR, CT, JBC), methodology (BFA, FR, CT, AEH, JMC, JBC), project administration (MR, SMN, AL, SPL, JMC), resources (SMN and AL), software (BFA, FR, CT), supervision (MR, SMN, AL, SPL, JMC), validation (BFA, FR, CT), visualization (BFA, FR), Writing—Original Draft Preparation (BFA), Writing—Review & Editing (BFA, FR, CT, SMN, MR, AE, AM, AJP, AL, CDA, KD, CPS, CN, SPL, JMC, AEH, JBC).

## Competing interests

The authors declare no competing interests.
