## [Peer Review File · Nature Communications]

Geographic pair matching in large-scale cluster randomized trialsREVIEWER COMMENTS

Reviewer #1 (Remarks to the Author):

The manuscript provides an empirical example that adds to the literature about a matched-pair design for large epidemiological trials, where geographic location is used to pair-match clusters (groups of individuals) before randomly allocating the paired clusters to one of two trial arms. The manuscript evaluates the benefits and limitations of using the matched-pair design for cluster randomised trials (CRTs) compared to other randomisation strategies (e.g., completely randomised design (simple) design), in terms of the relative statistical efficiency gains across multiple outcomes (primary and secondary outcomes). The empirical evaluation uses data from two large-scale cluster randomised controlled trials in Bangladesh and Kenya which evaluated interventions to improve nutrition and WASH (water, sanitation, hand washing).

The empirical evaluation is comprehensive, appropriate methods and techniques have been applied and generally the findings are clearly presented. The findings are applicable to specific large-scale public health trials, where geographical location, that encompasses socio-demographic characteristics of individuals and environmental factors) and are correlated with health outcomes of individuals (e.g, infectious diseases, nutrition), is used to pair/stratify the clusters before randomisation. Overall, the manuscript provides a balanced view and addresses the limitations of the evaluation. It also provides estimates of the outcome variance, matching correlation (albeit imprecise), intra-cluster-correlation (ICC) and Moran's I for a range of outcomes that can be used to inform future studies (Supp. Tables 1-3).

Some specific comments/questions:

1) Introduction: Although defined in the methods section, a suggestion is to explicitly state when describing the matched-pair (stratified) design that analysis needs to recognise the matching/stratification when assessing the relative statistical efficiency.

2) I found the interpretation in the section related to "Trial size and efficiency gains from geographic pair matching" (page 7-8) difficult to follow and may need further clarification.

What could explain the lack of relationship between trial size (number of paired clusters) and relative efficiency of a matched-pair design for outcomes that had smaller efficiency gains in the full sample? What role would the ICC of outcomes, and/or cluster size also have in increasing/decreasing the statistical efficiency of the matched-pair design?

For instance, I would expect that for outcomes with higher values of ICC (e.g. through greater geographic footprint by including more diverse populations / environmental characteristics), a matched-pair CRTs could potentially achieve higher statistical efficiency compared to outcomes where the between-cluster outcome variation is smaller (i.e. ICC approaches 0; keeping all other parameters constant). Given total variance is fixed, more effective matches could be induced that capture a greater degree of outcome variation between clusters (i.e., stronger matching correlation leading to increased statistical efficiency) when there is greater between-cluster variance (i.e. higher ICC). For instance, *Ascaris* sp. & EASQ gross motor z scores for Kenya trial had the strongest matching correlations & highest relative efficiency (Supp Table 3) but also highest ICC (Suppl 2). Increasing cluster size may also play a role achieving effective matching even when the ICC is approaching zero (See Table 2; DOI: 10.1002/sim.9152).

3) Please clarify whether the trial size refers to the “number of paired clusters” sampled, and cluster size is fixed?

4) Page 7-8, lines 283-303: Evaluation shows that matching on primary outcome at baseline showed greater gains in statistical efficiency for the primary outcome compared to matching on geographical location but matching on geographical location increased statistical efficiency across multiple outcomes. For trials that pair-match on geographic location, could there be additional gains (or not) in statistical efficiency when adjusting for the baseline measure of the primary outcome in the statistical analysis?

Similarly, could we increase statistical efficiency of a stratified design using geographic location (or completely randomised trial), and adjust for baseline measures correlated with the outcome in the analysis?

5) Discussion (Lines 442-443): Could another consideration for ensuring similar cluster sizes within pairs of clusters is that a design with balanced number of individuals between trial arms is more efficient than a trial where the sample size is not balanced between arms.

Lines 438-441: Would another caveat be that the two trials consisted of a relatively large number of paired clusters (n=90 and 72 matched pairs in Bangladesh and Kenya, respectively) for a range of outcomes specific to the context/setting. However, the findings may not be extrapolated for a trial with fewer than 10 pairs of clusters, and that the relative statistical efficiency may be less than 1 if the matching is not effective ($r < 0.11$) in this instance.

6) Page 16, Lines 687-688: Please clarify that the regression model used to estimate the ICC only includes the “control cluster” as random effect not the “pair”, as the analysis was restricted to control clusters.

Reviewer #2 (Remarks to the Author):

1. This paper is a large-scale simulation study to assess the gains in efficiency when clusters are matched according to their geographic distances. The merits to pair matching have been well-known to the literature over the past several years, so my understanding of the key contribution is to study the issue in two large-scale studies in epidemiology.

3. Strictly speaking, in the WASH studies the number of clusters in each tuples is not 2, so the procedure should be called matched tuples instead of pair matching.

4. 174–175: it appears each “pair” has 4 clusters, but in the previous paragraphs the number is said to be 8 or 9?

5. 188–189: What are the assumptions under which the relative efficiency boils down to the expression? Is it a superpopulation or a finite-population setting? Does the sampling framework accounts for pairing explicitly, as in the papers mentioned above, or assume pairs are drawn directly from the population, as in some other papers?

6. “We used group means in each geographically pair-matched block to estimate the pair-wise outcome correlation.” Mean of what exactly? There are two potential outcomes, i.e., potential outcomes if treated and not treated, for each unit in each cluster. These are

counterfactual outcomes, so how can one back out the unobserved counterfactuals from the data? Some assumption must be needed, like homogeneous treatment effects.

7. 206: What is an estimator that “ignored pairs”? Regardless of designs, as long as treated fractions stay constant across blocks or strata or pairs, we could always use the same estimator, the difference in means.

Reviewer #3 (Remarks to the Author):

Title:

Geographic pair-matching in large-scale cluster randomized trials (NCOMMS-23-18850)

Purpose and Rationale: The purpose of this manuscript is to present Geographical Pair-wise matching in the conduct of cluster -randomized trials as a strategy to improve the efficiency of trials with potential for cost savings. Authors indicate that in the event that cost is not saved the strategy is still able to get the best outcome in terms of power out of a trial.

Improvement in efficiency means that studies can be conducted with fewer participants without compromising the statistical precision when this approach is used.

This is based on the fact that individuals in a close geographic proximity share several important characteristics that may or may not be always measured in cluster randomized trials. To the extent that such characteristics affect outcomes, geographical pair matching becomes relevant.

Approach: Reanalysis of two cluster randomized large trials in Kenya and Bangladesh form the basis of this work. In conducting the re-analysis, the authors selected the control arm plus two arms that had a nutritional intervention.

Matching on geographical location (geographical pair-matching) is compared to two other strategies:

- a- Matching on a baseline variable that is closely related to an outcome (or several outcomes)
 - b- Stratified randomization where a small administrative unit such as a sub-district is used.
- Through the analysis however, the unmatched status is used for primary comparisons.

Results: In all analyses, geographical pair matching was found to improve efficiency

compared to unmatched trials. The central theme is that matching on geographical location is superior to unmatched design as well as to stratification at the level of small administrative units such as sub-districts.

The extent of improvement may be translated to practical benefits whereby an unmatched trial would have to enrol 2 – 3 times more clusters to achieve the same level of statistical precision. This is highly significant especially when a large expensive trial is considered.

Geographical pair matching was found to be superior to stratified approach in part due to the fact that the unit of stratification sometimes may be large enough to contain substantially variable characteristics. This is true in Kenya where even within a single subdistrict the population may differ quite significantly in their economic ability and health seeking behaviour.

Geographical pair matching was also found to have the advantage of dealing with heterogeneity in outcomes. However, for this part the mechanism is not entirely clear and would be difficult to replicate given the information provided.

Comment: Cluster randomized trials have great importance for health research in developing country settings especially in evaluating health interventions that are delivered at the group level (family/community, health facility etc). Efforts to improve the efficiency of cluster randomized trials without compromising the quality are important and this is clearly shown in this article.

The nature of the Kenyan populace is one where significant variability in culture, health seeking and economic ability exists first from one region to the next as well as within administrative units.

The fact that efficiency is improved for both large and fairly small cluster randomized trials is important for practical application. In practice the range of trials ranges from small to large.

Limitations: While the advantages of geographical pair matching are well presented in the paper, the limitations (if any) of this approach are not highlighted. For instance, it is important to clarify whether or not potential contamination between intervention and control is a concern when the clusters participating in a trial are extremely close. Secondly the pairwise geographical matching approach is mostly relevant for clusters within the community. It may not apply to situations where the clusters are for instance health units

because for very small administrative units it may just not be feasible (or it will not be very different from stratifying by administrative unit because the facilities may be so few).

Thirdly, in advancing the last advantage of geographical pair matching (heterogeneity) the message gets lost somewhat and this section would benefit from clearer examples.

Contribution to the field: This is an important paper that has potential to impact practice in the conduct of cluster randomized trials. It is well developed and clearly represents important advocacy for improving the efficiency of trials. Revision should address the points raised under limitations.

NCOMMS-23-18850A

Geographic pair matching in large-scale cluster randomized trials

REVIEWER COMMENTS

Reviewer #1 (Remarks to the Author):

The manuscript provides an empirical example that adds to the literature about a matched-pair design for large epidemiological trials, where geographic location is used to pair-match clusters (groups of individuals) before randomly allocating the paired clusters to one of two trial arms. The manuscript evaluates the benefits and limitations of using the matched-pair design for cluster randomised trials (CRTs) compared to other randomisation strategies (e.g., completely randomised design (simple) design), in terms of the relative statistical efficiency gains across multiple outcomes (primary and secondary outcomes). The empirical evaluation uses data from two large-scale cluster randomised controlled trials in Bangladesh and Kenya which evaluated interventions to improve nutrition and WASH (water, sanitation, hand washing).

The empirical evaluation is comprehensive, appropriate methods and techniques have been applied and generally the findings are clearly presented. The findings are applicable to specific large-scale public health trials, where geographical location, that encompasses socio-demographic characteristics of individuals and environmental factors) and are correlated with health outcomes of individuals (e.g, infectious diseases, nutrition), is used to pair/stratify the clusters before randomisation. Overall, the manuscript provides a balanced view and addresses the limitations of the evaluation. It also provides estimates of the outcome variance, matching correlation (albeit imprecise), intra-cluster-correlation (ICC) and Moran's I for a range of outcomes that can be used to inform future studies (Supp. Tables 1-3).

Response Thank you very much for this nice summary of the paper.

Some specific comments/questions:

1) Introduction: Although defined in the methods section, a suggestion is to explicitly state when describing the matched-pair (stratified) design that analysis needs to recognise the matching/stratification when assessing the relative statistical efficiency.

Response: We added this clarification in the second paragraph of the introduction, in two relevant places.

Lines 83-84

“Random allocation within strata or pair-matched clusters can improve statistical efficiency if the variable(s) used to stratify or match are strongly correlated with the outcome **and the stratification or matching variables are used in the analysis.**”

Lines 90-93

“The main benefits of pair-matching are that it ensures balance on characteristics used to match and that it can increase statistical power if trial outcomes are highly correlated within matched pairs **and the analysis accounts for the matched design.**”

2) I found the interpretation in the section related to “Trial size and efficiency gains from geographic pair matching” (page 7-8) difficult to follow and may need further clarification. What could explain the lack of relationship between trial size (number of paired clusters) and relative efficiency of a matched-pair design for outcomes that had smaller efficiency gains in the full sample? What role would the ICC of outcomes, and/or cluster size also have in increasing/decreasing the statistical efficiency of the matched-pair design?

For instance, I would expect that for outcomes with higher values of ICC (e.g. through greater geographic footprint by including more diverse populations / environmental characteristics), a matched-pair CRTs could potentially achieve higher statistical efficiency compared to outcomes where the between-cluster outcome variation is smaller (i.e. ICC approaches 0; keeping all other parameters constant).

Given total variance is fixed, more effective matches could be induced that capture a greater degree of outcome variation between clusters (i.e., stronger matching correlation leading to increased statistical efficiency) when there is greater between-cluster variance (i.e. higher ICC).

For instance, Ascaris sp. & EASQ gross motor z scores for Kenya trial had the strongest matching correlations & highest relative efficiency (Supp Table 3) but also highest ICC (Suppl 2). Increasing cluster size may also play a role achieving effective matching even when the ICC is approaching zero (See Table 2; DOI: 10.1002/sim.9152).

Response: In response to these helpful suggestions, the revised manuscript includes additional clarification in the results that describe the relationship between trial size and gains in statistical efficiency. We replaced the equivocal interpretation in the original draft with this more clear connection with measures between cluster and spatial variance suggested by the reviewer. We also included in the Discussion a new paragraph on generalizability of the findings, suggested by Reviewer 3, which includes this suggestion regarding the influence of cluster size and pair matching gains for outcomes with low ICC. We added a citation to this very nice paper suggested by the reviewer that reports simulation results that align with our empirical findings.

Lines 261-271 (Results)

“Outcomes with greatest gains in relative efficiency at larger trial sizes were those with greater between cluster variance and spatial autocorrelation. For example, in Bangladesh outcomes with highest gains in relative efficiency had high ICCs: *Ascaris* sp. (ICC=0.05), *Giardia* sp. (0.07), length-for-age z (0.06), weight-for-age z (0.07) and *Trichuris* sp. (0.05) and similarly high values of Moran’s I (Supplementary Table 1). Patterns were similar for Kenya (Supplementary Table 2) and reinforce the positive relationship between efficiency gains and between-cluster variability and spatial autocorrelation (Supplementary Fig. 5). Given total variance is fixed, the results suggest more effective geographic matches capture a greater degree of outcome variation between clusters when there is greater between-cluster variance and spatial autocorrelation in the outcome.”

Lines 508-511 (Discussion)

“Finally, the relationship between efficiency gains and measures of intra-cluster correlation and spatial autocorrelation could be different in trials with larger clusters — a recent simulation study showed that increasing cluster size may play a role achieving effective matching even when the ICC approaches zero.³⁶”

3) Please clarify whether the trial size refers to the “number of paired clusters” sampled, and cluster size is fixed?

Response: The reviewer has correctly inferred our approach. We clarified this point in the text. We also ensured this was clear in the Figure 3 axis label, which is now “Down-sampled trial size: number of geographically proximate matched pairs”

Lines 250-251 (Results)

“The resampling approach held cluster sizes fixed at their actual size, so resampled trials varied in size according to the number of matched pairs.”

Lines 733-734 (Methods)

“Clusters were kept at their observed sizes, and so the different subsampled trials varied in size according to the number of matched pairs.”

4) Page 7-8, lines 283-303: Evaluation shows that matching on primary outcome at baseline showed greater gains in statistical efficiency for the primary outcome compared to matching on geographical location but matching on geographical location increased statistical efficiency across multiple outcomes. For trials that pair-match on geographic location, could there be additional gains (or not) in statistical efficiency when adjusting for the baseline measure of the primary outcome in the statistical analysis?

Similarly, could we increase statistical efficiency of a stratified design using geographic location (or completely randomised trial), and adjust for baseline measures correlated with the outcome in the analysis?

Response: This is a very interesting suggestion. Since these particular trials could not measure baseline outcomes (children were not yet born) we could not formally assess this here, but we added this to the caveats of the Discussion and included a new citation that elaborates on additional details of a combined strategy of matching and further regression adjustment.

Lines 476-482 (Discussion)

"The trials did not have baseline outcome measurements as children were not yet born so we could not assess whether additional gains might be possible by combining geographic pair matched (or stratified) randomization with additional regression adjustment for baseline outcomes. Additional gains may be possible by controlling for residual outcome variation not captured in the matched or stratified randomization, but care must be taken in selection of adjustment covariates and the analysis methods add complexity compared with the unadjusted estimators used here.³²"

5) Discussion (Lines 442-443): Could another consideration for ensuring similar cluster sizes within pairs of clusters is that a design with balanced number of individuals between trial arms is more efficient than a trial where the sample size is not balanced between arms.

Response: Thank you for this suggestion. We added this point to the Discussion

Lines 470-472 (Discussion)

"Similar cluster sizes within pairs also leads to a balanced number of individuals in each group, which is more statistically efficient compared to an imbalanced design."

Lines 438-441: Would another caveat be that the two trials consisted of a relatively large number of paired clusters (n=90 and 72 matched pairs in Bangladesh and Kenya, respectively) for a range of outcomes specific to the context/setting. However, the findings may not be extrapolated for a trial with fewer than 10 pairs of clusters, and that the relative statistical efficiency may be less than 1 if the matching is not effective ($r < 0.11$) in this instance.

Response: This is a helpful suggestion, and we added this additional clarification on the generalizability of the results to the Discussion along with a key point made by Reviewer 3 regarding the nature of "clusters" studied in these trials versus other types of clusters, such as healthcare facilities. Although the focus of the paper is clearly on large-scale trials (as

reinforced in the title and the introduction), this serves as a helpful reminder for readers in the later section of the paper.

Lines 494-498

"When considering the generalizability of the findings, note that the results arise from two community-based trials with a relatively large number of pairs, and outcomes that were specific to the rural Bangladesh and Kenya contexts. The results should not be extrapolated to trials with fewer than 10 clusters where the relative efficiency could be worse if geographic pair matching induces a weak correlation in outcomes ($r < 0.11$)."

6) Page 16, Lines 687-688: Please clarify that the regression model used to estimate the ICC only includes the "control cluster" as random effect not the "pair", as the analysis was restricted to control clusters.

Response: Thank you for catching this typo. We corrected the text to replace pair with cluster.

Line 746-747

"We fit an intercept-only linear regression model on individual-level outcomes that included a random effect for **cluster**."

Reviewer #2 (Remarks to the Author):

1. This paper is a large-scale simulation study to assess the gains in efficiency when clusters are matched according to their geographic distances. The merits to pair matching have been well-known to the literature over the past several years, so my understanding of the key contribution is to study the issue in two large-scale studies in epidemiology.

Response: We agree that the theoretical merits of pair matching have been described in the past several years, particularly in the statistical literature. Part of the real-world challenge with pair matching in general is that studies often lack baseline data with which to match. As we discuss in the introduction, a major advantage of geographic pair matching in large-scale trials is that geographic location is almost always known at the time of randomization, so matching by geography is logistically feasible and imposes very little cost to the trial. To our knowledge, there have been no estimates of merits of geographic pair matching and this study represents a broad assessment of efficiency gains across diverse child outcomes in two different settings. Another novel contribution is the approach we proposed to estimate effect heterogeneity by spatially varying covariates, which is uniquely simple in a pair matched design but has not been previously described. This is a minor clarification, but we do not view this paper as a simulation study since all estimates rely on primary data

collected in the trials (not derived through simulation). We anticipate the novel empirical results reported in this paper, presented to a more general scientific audience (rather than a purely statistical audience), might increase awareness among the research community of the potential benefits of geographic pair matching in the design of future large-scale trials.

3. Strictly speaking, in the WASH studies the number of clusters in each tuples is not 2, so the procedure should be called matched tuples instead of pair matching.

Response: Thank you for pointing out this correct nomenclature. We clarified this in the results.

Lines 169-172

"The trials technically matched 8-tuples but for clarity of exposition in the present analysis, we limited the study population to control and nutritional intervention clusters (N, WSH+N), and pooled outcomes within each group to create a simplified, balanced design with which to focus on the methodologic aspects of geographic pair matching (Fig. 1c)."

4. 174–175: it appears each "pair" has 4 clusters, but in the previous paragraphs the number is said to be 8 or 9?

Response: Please see our response to the previous comment to clarify this.

5. 188–189: What are the assumptions under which the relative efficiency boils down to the expression? Is it a superpopulation or a finite-population setting? Does the sampling framework accounts for pairing explicitly, as in the papers mentioned above, or assume pairs are drawn directly from the population, as in some other papers?

Response: The reviewer has astutely noticed that we provided the simplest expression of relative efficiency in the brief introduction to the results, with a parenthetical "(details in Methods)" To keep the results concise we qualified that this expression is the relationship "in its simplest form". In the Methods, we clarified the basic assumptions that underlie the analyses and referred readers to previous work that includes formal details.

Lines 606-615

"Formal statistical assumptions of pair matched cluster randomized trials have been defined.¹⁵ In brief, we assume that treatment is randomized at the cluster level and that potential outcomes of individuals within clusters do not depend on the treatment status of other clusters, including matched pair clusters. The second assumption (no contamination or spillover between clusters) is reasonable in this context as clusters were separated by ≥ 1 km to prevent spillover, and an absence of between-cluster spillover has been previously demonstrated for several outcomes in the trials.^{4,5,35} Finally, we assume that clusters represent a random sample from the larger population but all individuals within a cluster are

observed. This reflects the trials' design since clusters consisted of all eligible mothers and their newborn children and has been referred to as the unit average treatment effect.¹⁵

6. "We used group means in each geographically pair-matched block to estimate the pair-wise outcome correlation." Mean of what exactly? There are two potential outcomes, i.e., potential outcomes if treated and not treated, for each unit in each cluster. These are counterfactual outcomes, so how can one back out the unobserved counterfactuals from the data? Some assumption must be needed, like homogeneous treatment effects.

Response: We have clarified that we used cluster level means in each matched pair to estimate pair-wise outcome correlation.

The reviewer's suggestion is correct that a measure of outcome correlation between matched pairs will include the underlying outcome correlation plus some influence of the treatment effect — whether it be homogeneous or not. As we show in Fig 2b, this approximation turns out to be quite good for all outcomes in the two trials, albeit the observed, empirical efficiency gains are slightly larger than gains predicted by correlation alone. We clarified this point in the text:

Lines 198-201 (Results)

"We estimated mean outcomes at the cluster level, and then estimated the correlation between cluster-level outcomes within matched pairs, noting that observed outcomes reflect correlation (if any) induced by matching as well as the treatment effect, which may be heterogeneous across pairs."

7. 206: What is an estimator that "ignored pairs"? Regardless of designs, as long as treated fractions stay constant across blocks or strata or pairs, we could always use the same estimator, the difference in means.

Response: We see how this could be ambiguous with the Results section coming before the Methods for this paper. We included more specific language here. The reviewer is correct, we were referring to the difference in means.

Lines 211-213 (Results)

"Observed efficiency gains that compared the variance of the pair matched estimator with a **difference in means** estimator that ignored **pair matching** were very close, albeit slightly larger, than gains predicted based on weighted correlation of pair-wise outcomes (Fig. 2b),"

(note: in the methods, the estimator that did not account for matching, "ignored pairs", is defined very specifically, but we did not include this detailed notation in the results for

parsimony, i.e. this is the text from the Methods, : In a study with a total of n children and c clusters, and n_j children in cluster j , we defined the unmatched ATE for potential outcomes of child i in cluster j as: $\psi_{ATE}^U = \frac{1}{n} \sum_{j=1}^c \sum_{i=1}^{n_j} (Y_{1ij} - Y_{0ij})$. The non-parametric estimator is:

$$\hat{\psi}_{ATE}^U = \frac{1}{n} \sum_{j=1}^c \sum_{i=1}^{n_j} [A_j Y_{ij} - (1 - A_j) Y_{ij}]$$

Reviewer #3 (Remarks to the Author):

Title:

Geographic pair-matching in large-scale cluster randomized trials (NCOMMS-23-18850)

Purpose and Rationale: The purpose of this manuscript is to present Geographical Pair-wise matching in the conduct of cluster -randomized trials as a strategy to improve the efficiency of trials with potential for cost savings. Authors indicate that in the event that cost is not saved the strategy is still able to get the best outcome in terms of power out of a trial. Improvement in efficiency means that studies can be conducted with fewer participants without compromising the statistical precision when this approach is used.

This is based on the fact that individuals in a close geographic proximity share several important characteristics that may or may not be always measured in cluster randomized trials. To the extent that such characteristics affect outcomes, geographical pair matching becomes relevant.

Approach: Reanalysis of two cluster randomized large trials in Kenya and Bangladesh form the basis of this work. In conducting the re-analysis, the authors selected the control arm plus two arms that had a nutritional intervention.

Matching on geographical location (geographical pair-matching) is compared to two other strategies:

a- Matching on a baseline variable that is closely related to an outcome (or several outcomes)

b- Stratified randomization where a small administrative unit such as a sub-district is used.

Through the analysis however, the unmatched status is used for primary comparisons.

Response: Thank you for this accurate summary of the rationale and approach.

Results: In all analyses, geographical pair matching was found to improve efficiency compared to unmatched trials. The central theme is that matching on geographical location is superior to unmatched design as well as to stratification at the level of

small administrative units such as sub-districts.

The extent of improvement may be translated to practical benefits whereby an unmatched trial would have to enrol 2 – 3 times more clusters to achieve the same level of statistical precision. This is highly significant especially when a large expensive trial is considered.

Geographical pair matching was found to be superior to stratified approach in part due to the fact that the unit of stratification sometimes may be large enough to contain substantially variable characteristics. This is true in Kenya where even within a single subdistrict the population may differ quite significantly in their economic ability and health seeking behaviour.

Geographical pair matching was also found to have the advantage of dealing with heterogeneity in outcomes. However, for this part the mechanism is not entirely clear and would be difficult to replicate given the information provided.

Response: This summary is accurate, but we would characterize the final point slightly differently, which is that geographic pair matching enables a trial to assess spatial heterogeneity in the intervention effect and/or effect modification of the intervention by spatially varying covariates using relatively simple methods. We have described the approach in detail and have provided the underlying computer code to replicate to provide the most visibility possible to readers. Please see our response below to this reviewer's three primary concerns for additional responses related to this suggestion. The reviewer has made a good point that we did not discuss or study the mechanism underlying the observed spatial heterogeneity. In the Discussion, we noted the potential analytic extensions to study mechanism for spatial heterogeneity:

Lines 450-454

"Beyond population remoteness, we did not study mechanisms that underlie the observed spatial effect heterogeneity in Bangladesh. Since geography integrates many environmental and socioeconomic characteristics, the mechanism likely multifactorial but such analytic extensions could provide useful information for future intervention planning."

Comment: Cluster randomized trials have great importance for health research in developing country settings especially in evaluating health interventions that are delivered at the group level (family/community, health facility etc). Efforts to improve the efficiency of cluster randomized trials without compromising the quality are important and this is clearly shown in this article.

The nature of the Kenyan populace is one where significant variability in culture, health seeking and economic ability exists first from one region to the next as well as within administrative units.

The fact that efficiency is improved for both large and fairly small cluster randomized trials is important for practical application. In practice the range of trials ranges from small to large.

Limitations: While the advantages of geographical pair matching are well presented in the paper, the limitations (if any) of this approach are not highlighted. For instance, it is important to clarify whether or not potential contamination between intervention and control is a concern when the clusters participating in a trial are extremely close. Secondly the pairwise geographical matching approach is mostly relevant for clusters within the community. It may not apply to situations where the clusters are for instance health units because for very small administrative units it may just not be feasible (or it will not be very different from stratifying by administrative unit because the facilities may be so few). Thirdly, in advancing the last advantage of geographical pair matching (heterogeneity) the message gets lost somewhat and this section would benefit from clearer examples.

Contribution to the field: This is an important paper that has potential to impact practice in the conduct of cluster randomized trials. It is well developed and clearly represents important advocacy for improving the efficiency of trials. Revision should address the points raised under limitations.

Response: We are grateful to the reviewer for these helpful suggestions for clarification that have, in our view, substantially improved the Discussion section of the paper. The reviewer raises three key points that we will address in turn. The first two points inspired much of the material for a new paragraph on the generalizability and limitations of the analysis. To make it easier to see the context, we have provided the entire paragraph below, but have highlighted the most relevant sections using bold font for points 1 and 2.

1. **Contamination between clusters.** This is an important point. We have addressed it both in general and in the specific context of this trial. See response to point 2 for the revised text in the Discussion.
2. **Generalizability to trials that randomize units other than communities, such as healthcare centers.** This is an important insight from the reviewer, which we had not previously considered. We added this important point to a new paragraph that discusses the generalizability of the study's findings:

Lines 494-511 (Discussion)

"When considering the generalizability of the findings, note that the results arise from two community-based trials with a relatively large number of pairs, small cluster sizes, and outcomes that were specific to the rural Bangladesh and Kenya contexts. The results should not be extrapolated to trials with fewer than 10 clusters where the relative efficiency could be worse if geographic pair matching induces a weak correlation in outcomes ($r < 0.11$). **Moreover, it is unclear whether geographic pair matching of randomized units other than communities, such as healthcare**

centers, would obtain similar gains in efficiency or could be used to study fine-scale spatial heterogeneity in treatment effects as we have done here. Matched clusters were geographically close in the trials (~1 km spacing) and so matched pairs separated by larger geographic distances might benefit less from pair matching depending on the scale of geographic variation in the outcome. As with unmatched cluster randomized trials, a key assumption of matched pair trials is no interference (no spillover) between matched pairs.¹⁵ Although clusters were relatively close in both trials, they were separated by ≥ 1 km to prevent contamination between them and empirical estimates in the original trials demonstrated an absence of spillover effects.^{4,5,35} Finally, the relationship between efficiency gains and measures of intra-cluster correlation and spatial autocorrelation could be different in trials with larger clusters — a recent simulation study showed that increasing cluster size may play a role achieving effective matching even when the ICC approaches zero.³⁶

- 3. Need for additional clarification in examples that demonstrate methods to study effect heterogeneity by taking advantage of the pair matched design.** We appreciate this suggestion to improve the exposition for this part of the analysis. In the revision, we made several additions to the Results section to help make the motivation for the approach and the methods used clearer in the initial presentation of the material (i.e., before the Methods). We made these changes in the revised manuscript Results section, lines 322-361.

(not including here b/c several changes — they are tracked in revised manuscript)

REVIEWERS' COMMENTS

Reviewer #1 (Remarks to the Author):

The authors have appropriately addressed my comments and queries. I have no further comments.

Reviewer #3 (Remarks to the Author):

The revised version of the manuscript has addressed comprehensively the comments and suggestion provided during the initial review.

Specific to my comments the authors have addressed the points of contamination, generalizability which are important aspects to the scientific community that will utilize the knowledge gained from this paper.

Reviewer #4 (Remarks to the Author):

This is a useful and well written re-analysis of cluster-randomised trial data to demonstrate potential gains through geographical pair matching of clusters. This uses data from two trials in Kenya and Bangladesh evaluating nutrition and WASH interventions to improve multiple indicators of child health. Overall, the results are clearly presented and illustrate the increased efficiency of geographical pair matching of cluster randomised trials in comparison to simple randomization or matching on a baseline variable. The authors additionally present useful analysis of the varying efficacy of interventions based on a spatially varying covariate, which makes sense logically considering the types of interventions assessed and is useful to quantify.

I did not review previous versions of this paper and, from the provided comments, the authors have effectively responded to most comments. However, I believe this analysis and manuscript would benefit from additional discussion of the limitations of this approach and how spatially correlated health processes may interact with effect estimates. The authors state that there were no spillover effects as the clusters were >1km apart. For many of the outcomes assessed, such as infectious disease transmission or information provided to

mothers, it is not unreasonable to assume spillover effects within such a distance as there is a high probability these individuals would come into contact. It would be helpful to understand more on how the possibility of spillover effects were ruled out. Depending on the processes to assess, this is likely one of the major limitations of a geographic pair-matching approach.

In general, it would be useful to identify conditions where this approach is likely appropriate, i.e. based on the processes assessed or distribution of the community, and how this approach may be impacted by the spatial structure of these processes. For example, there may be minimal benefit to geographical pair matching for a highly focal outcome while this may be more suitable for spatially correlated outcomes with larger spatial ranges. While it may not be possible with these trial datasets, it would be helpful to see if there were any baseline estimates of spatial correlation for different outcomes and how this impacted the results. It is likely that identifying how to geographically pair match clusters (what distance to match, how to rule out contamination) would be one of the key design questions for practitioners intending to use this method.

Overall, I would recommend this manuscript for publication with minor edits.

NCOMMS-23-18850B

Geographic pair matching in large-scale cluster randomized trials

REVIEWER COMMENTS

Reviewer #1 (Remarks to the Author):

The authors have appropriately addressed my comments and queries. I have no further comments.

Response Thank you very much for your constructive reviews.

Reviewer #3 (Remarks to the Author):

The revised version of the manuscript has addressed comprehensively the comments and suggestion provided during the initial review.

Specific to my comments the authors have addressed the points of contamination, generalizability which are important aspects to the scientific community that will utilize the knowledge gained from this paper.

Response Thank you very much for your constructive reviews and for confirming each major point has been addressed satisfactorily.

Reviewer #4 (Remarks to the Author):

This is a useful and well written re-analysis of cluster-randomised trial data to demonstrate potential gains through geographical pair matching of clusters. This uses data from two trials in Kenya and Bangladesh evaluating nutrition and WASH interventions to improve multiple indicators of child health. Overall, the results are clearly presented and illustrate the increased efficiency of geographical pair matching of cluster randomised trials in comparison to simple randomization or matching on a baseline variable. The authors additionally present useful analysis of the varying efficacy of interventions based on a spatially varying covariate, which makes sense logically considering the types of interventions assessed and is useful to quantify.

I did not review previous versions of this paper and, from the provided comments, the authors have effectively responded to most comments. However, I believe this analysis and manuscript would benefit from additional discussion of the limitations of this approach and how spatially correlated health processes may interact with effect

estimates. The authors state that there were no spillover effects as the clusters were >1km apart. For many of the outcomes assessed, such as infectious disease transmission or information provided to mothers, it is not unreasonable to assume spillover effects within such a distance as there is a high probability these individuals would come into contact. It would be helpful to understand more on how the possibility of spillover effects were ruled out. Depending on the processes to assess, this is likely one of the major limitations of a geographic pair-matching approach.

Response: The reviewer makes a very reasonable observation. In theory, we completely agree with the potential for spillover effects between clusters for both infectious disease outcomes and for informational components of the intervention that could be passed to nearby communities through word of mouth. Indeed, we have completed multiple analyses of the trials in this regard under the hypothesis that there could be spillover effects between clusters. All analyses have demonstrated no spillover.

The basic approach we used to test for spillover effects was to use a permutation test to identify any effects on trial outcomes in control clusters as a function of distance to intervention clusters or through shared attendance at institutions (schools, markets, churches/mosques). The details of the statistical approach are in the original trials' statistical analysis plan (available here, starting on p4: <https://osf.io/kw3tp>) and were reported in the primary outcome papers' supplemental materials. There was no evidence for spillover effects on diarrhea or growth.

<https://pubmed.ncbi.nlm.nih.gov/29396217/>

<https://pubmed.ncbi.nlm.nih.gov/29396219/>

Additionally, in the Bangladesh trial, we completed a spillover substudy focused on measuring infectious disease, environmental contamination, and behavioral outcomes among compounds that were within study communities, adjacent to enrolled compounds but not enrolled in the trial itself. Like the reviewer, we hypothesized that the intervention could influence outcomes or behaviors in people living nearby but not enrolled in the trial. However, we found no evidence of spillover effects on any outcome, even at this very short distance (far closer than the 1km distance between trial clusters). Additional details here:

<https://pubmed.ncbi.nlm.nih.gov/29596644/>

We have clarified this point in the Discussion. Please see the response to the next comment for a summary of changes.

In general, it would be useful to identify conditions where this approach is likely appropriate, i.e. based on the processes assessed or distribution of the community, and how this approach may be impacted by the spatial structure of these processes. For example, there may be minimal benefit to geographical pair matching for a highly focal outcome while this may be more suitable for spatially correlated outcomes with

larger spatial ranges. While it may not be possible with these trial datasets, it would be helpful to see if there were any baseline estimates of spatial correlation for different outcomes and how this impacted the results. It is likely that identifying how to geographically pair match clusters (what distance to match, how to rule out contamination) would be one of the key design questions for practitioners intending to use this method.

Overall, I would recommend this manuscript for publication with minor edits.

Response: Thank you for this interesting observation. We agree. The trials were unusual in that they did not include baseline measurements of child health outcomes since the children were in utero at the time of enrollment and randomization. However, we can approximate outcome spatial correlation using measurements in control clusters, under an assumption of no spillover effects (which we believe is reasonable, per our response to the previous comment). Inspired by the reviewer's suggestion, we conducted an additional analysis that summarized the spatial correlation for the outcomes in the two trials, which we summarized in new Supplementary Figures 9 and 10. We found that outcome correlation decays rapidly to zero by 10-20 km for nearly all outcomes, showing that the outcome correlation appeared to be highly focal in these studies. We added this information to the results section in the context of explaining why a pair matched design would be more statistically efficient than a subdistrict stratified design.

We also included a new paragraph in the Discussion that more clearly describes the issue the reviewer has raised about a potential trade-off in terms of geographic proximity of matched clusters vs. the potential for spillover between them. We included the previous material related to spillover effects in that paragraph (moved up from a later paragraph).

Below are the sections of revised text, with changes indicated in **bold font**.

Results

"The subdistrict stratified analysis improved efficiency for most outcomes compared to an unadjusted analysis, but for nearly every outcome the pair matched estimator had the lowest variance (highest relative efficiency) — in many cases, the subdistrict stratified analysis was substantially less efficient compared with the pair matched analysis (Fig. 4). For five outcomes in Bangladesh, all efficiency gains were lost with subdistrict stratification. These results show that fine stratification by geography obtained through pair matching has potential to further improve efficiency compared with a design that stratifies on small administrative units, which implies there is substantial outcome variation at spatial scales below subdistrict. **To further interrogate this result, we estimated spatial outcome correlation by distance using a universal kriging model with a semiparametric smooth of control cluster means (Methods). Spatial outcome correlation fell to zero by 10-20**

km for nearly all outcomes (Supplementary Figs. 9,10), demonstrating that spatial correlation was generally below subdistrict scale. "

Discussion

"Future studies could also assess optimal distance between matched pairs, which should depend on spatial outcome correlation patterns and the potential for spillover between clusters. Spatial outcome correlation disappeared by 10-20 km for almost all outcomes studied here (Supplementary Figs. 9,10), suggesting that geographically close pairing (~ 1 km) was beneficial for efficiency gains. Matched pairs separated by larger geographic distances might benefit less from pair matching depending on the scale of geographic variation in the outcome. As with unmatched cluster randomized trials, a key assumption of matched pair trials is no interference (spillover) between matched pairs.¹⁵ Although clusters were relatively close in both trials, they were separated by ≥ 1 km to prevent contamination between them and empirical estimates in the original trials demonstrated an absence of spillover effects **on infectious disease, child growth, measures of environmental contamination and behavioral outcomes.**^{4,5,31"}